# Diet-induced adaptive thermogenesis requires neuropeptide FF receptor-2 signalling

Lei Zhang [1,2], Chi Kin Ip[1], I-Chieh J. Lee[1], Yue Qi[1], Felicia Reed[1], Tim Karl[3,4,5], Jac Kee Low [4], Ronaldo F. Enriquez[1,6], Nicola J. Lee[1,2], Paul A. Baldock[5,6,7] & Herbert Herzog [1,5,7]

Excess caloric intake results in increased fat accumulation and an increase in energy expenditure via diet-induced adaptive thermogenesis; however, the underlying mechanisms controlling these processes are unclear. Here we identify the neuropeptide FF receptor-2 (NPFFR2) as a critical regulator of diet-induced thermogenesis and bone homoeostasis. *Npffr2*$^{-/-}$ mice exhibit a stronger bone phenotype and when fed a HFD display exacerbated obesity associated with a failure in activating brown adipose tissue (BAT) thermogenic response to energy excess, whereas the activation of cold-induced BAT thermogenesis is unaffected. NPFFR2 signalling is required to maintain basal arcuate nucleus NPY mRNA expression. Lack of NPFFR2 signalling leads to a decrease in BAT thermogenesis under HFD conditions with significantly lower UCP-1 and PGC-1α levels in the BAT. Together, these data demonstrate that NPFFR2 signalling promotes diet-induced thermogenesis via a novel hypothalamic NPY-dependent circuitry thereby coupling energy homoeostasis with energy partitioning to adipose and bone tissue.

[1] Neuroscience Division, Garvan Institute of Medical Research, St. Vincent's Hospital, Darlinghurst, NSW, Australia. [2] St. Vincent's Clinical School, University of NSW, Sydney, NSW, Australia. [3] School of Medicine, Western Sydney University, Sydney, NSW, Australia. [4] Neuroscience Research Australia, Randwick, NSW, Australia. [5] School of Medical Sciences, University of NSW, Sydney, NSW, Australia. [6] Bone Biology Division, Garvan Institute of Medical Research, St Vincent's Hospital, Darlinghurst, NSW, Australia. [7] Faculty of Medicine, University of NSW, Sydney, NSW, Australia. Correspondence and requests for materials should be addressed to H.H. (email: h.herzog@garvan.org.au)

Energy homoeostasis is fundamental for survival and there are adaptive mechanisms in place to counteract energy imbalances[1,2]. In particular, under a positive-energy balance induced by excessive caloric intake, energy expenditure is increased to limit weight gain[3–5]. This capacity to increase energy expenditure in response to caloric excess is commonly referred to as diet-induced thermogenesis and consistently observed in humans[4,6] and rodents[3,5]. One of the most critical tissues facilitating this process is brown adipose tissue (BAT), which, through an uncoupling protein-1 (UCP-1)-dependent mechanism, activates diet-induced thermogenesis[7,8]. Recent studies also revealed a creatine-futile-cycle pathway in white adipose tissue contributing to this thermogenetic response[9]. However, in comparison with the research advances made in this area in the periphery, far less is known about the central mechanisms.

The brain receives a variety of peripheral signals conveying nutrient status, thereby detecting perturbations in energy balance and allowing for the activation of adaptive mechanisms. The hypothalamus and brain stem are two key regions where peripheral circulating factors and sensory afferent fibres integrate with the central circuitry with a variety of different neuropeptide-expressing neuronal populations involved[10–12]. Among them is a group of neuropeptides belonging to the RF-amide peptide family including the neuropeptide FF group (NPFF and neuropeptide AF), the RFamide-related peptides group (RFRP-1 and -3, also named NPSF and NPVF), the prolactin-releasing peptide (PrRP) group, the pyroglutamylated RFamide peptide group (26RF and QRFP) and the kisspeptin group[13]. Interestingly, physiological studies have linked them to the regulation of feeding and energy homoeostasis[14]. For example, NPFF, which is expressed by neurons in the brain stem[15,16], has been shown to reduce short-term feeding in rats and in pre-fasted and ad libitum-fed mice[17,18]. PrRP-expressing neurons expressed in the brain stem have been suggested to relay signals that mediate satiety[17], whereas PrRP-expressing neurons in the dorsal medial hypothalamus are thought to mediate the thermogenic effect of leptin[19]. The kisspeptin system — best known for its role in stimulating the reproductive axis — is also increasingly recognized as an important factor in the energy metabolism and glucose regulation[20,21].

RFamide peptides exert their functions via $G_{i/o}$ associated G-protein-coupled receptors of which five have been identified in mammals, i.e., NPFFR1, NPFFR2, GPR10, GPR103 and KISS1R[13]. Pharmacological studies have revealed that the endogenous ligands NPFF and RFRPs are recognized by more than one receptor adding to the complexity of this system[22]. NPFFR2, which can be activated by NPFF, NPVF, PrRP and kisspeptin[22–24], has the highest levels of expression in the hypothalamic nuclei that are linked to the regulation of energy homoeostasis[25]. Thus, it is likely to be that NPFF, NPVF, PrRP and kisspeptin influence aspects of energy homoeostasis predominantly via signalling at the NPFFR2.

Multi-ligand/multi-receptor systems with their susceptibility to cross-reactions impose great challenges on investigations into the specificities of the peptide and receptor functions. This is further complicated by the lack of selective agonist or antagonist suitable for in vivo investigation. Thus, we generated a variety of NPFFR2 mouse models and systematically examined the impact of altering NPFFR2 signalling on parameters of energy balance and its potential involvement in the regulation of diet-induced thermogenesis. Our results show that NPFFR2 signalling in the arcuate nucleus of the hypothalamus is crucial in maintaining a basal tone of neuropeptide Y (NPY), lack of which renders downstream pathways that control adaptive thermogenesis in the BAT unresponsive to caloric excess and consequently leads to exacerbated diet-induced obesity.

## Results

**Energy status regulates the expression of NPFFR2 ligands.** The expression of the main ligands of NPFFR2 (NPFF, NPVF and PrRP) are altered depending on energy status with $Npff$ mRNA being significantly reduced in response to 8 weeks of high-fat diet (HFD) feeding (Fig. 1a–c) and significantly increased after 24 h fasting (Fig. 1c). The hypothalamic expression of $Npvf$ was significantly reduced in response to HFD (Fig. 1d) but unaltered by fasting (Fig. 1d). PrRP ($Prlh$) mRNA levels were significantly increased in the nucleus tractus solitarius (NTS; Fig. 1e, f) but were unaltered in the hypothalamus in response to HFD feeding (Fig. 1g), whereas hypothalamic $Prlh$ mRNA expression was strongly reduced in response to fasting (Fig. 1g).

To test whether the expression of these ligands is under the control of leptin, we examined subpostrema area (SubP) $Npff$ and NTS $Prlh$ mRNA expression in $Lep^{-/-}$ mice. Consistent with the downregulation of SubP $Npff$ and upregulation of NTS $Prlh$ expression under HFD feeding (high leptin), $Lep^{-/-}$ mice exhibited significantly increased and decreased expression of SubP N$pff$ and NTS $Prlh$, respectively (Fig. 1h–j). To examine whether leptin has a direct action on $Npff$ expression, we conducted dual in situ hybridization (ISH) for $Npff$ and $Lepr$. Although $Npff$ staining is elevated in $Lep^{-/-}$ mice (Fig. 1k) — consistent with Fig. 1h — little or no colocalization of $Lepr$ and $Npff$ was evident (Fig. 1k), suggesting an indirect action of leptin, if any, in the regulation of $Npff$ mRNA expression.

**Generation and behavioural examination of $Npffr2^{-/-}$ mice.** To investigate the role of NPFFR2 signalling in the regulation of energy homoeostasis, we generated germline $Npffr2^{-/-}$ and conditional $Npffr2^{lox/lox}$ mouse models (Supplementary Fig. 1). Breeding of heterozygous $Npffr2^{+/-}$ mice resulted in live offspring with a frequency of genotypes that was not significantly different from the expected Mendelian ratio (Supplementary Fig. 2a) and with litter sizes actually being larger than those from wild-type (WT) breeding pairs (Supplementary Fig. 2b). Importantly, homozygous $Npffr2^{-/-}$ breeding pairs produced litters of a similar size to those of WT breeding pairs (Supplementary Fig. 2b), indicating that deletion of the $Npffr2$ does not affect fertility. Mortality rate and gender ratio of pups born from heterozygous and homozygous breeding pairs were not significantly different from those of WT (Supplementary Fig. 2c–d), suggesting that disruption of $Npffr2$ expression does not impact upon reproductive fitness.

$Npffr2^{-/-}$ and littermate control mice were further tested in behavioural test paradigms to identify potential abnormalities that could influence energy homoeostasis. $Npffr2^{-/-}$ mice of both genders did not differ from controls with regard to motor function or sensorimotor gating (Supplementary Table 1). In addition, neither the open-field test nor the elevated plus maze test revealed an anxiety-related phenotype (Supplementary Table 1). Finally in a hot water tail-flick test, the latency to flick the tail was comparable between $Npffr2^{-/-}$ and WT mice (Supplementary Table 1), indicating an unaltered responsiveness of $Npffr2^{-/-}$ mice to a thermal stimulus.

**Enhanced diet-induced obesity in $Npffr2^{-/-}$ mice.** On a standard chow diet, male $Npffr2^{-/-}$ mice displayed a similar growth curve as WT mice (Fig. 2a). However, weight gain over a 10-week monitoring period (i.e., between 10 and 20 weeks of age) showed a small but significant reduction in the $Npffr2^{-/-}$ mice (Fig. 2b). Whole-body lean and fat masses assessed by dual-energy X-ray absorptiometry (DEXA) (Supplementary Fig. 3a, b) and adiposity assessed by the weights of dissected white adipose tissue depots (Fig. 2c, d) were unaltered in male $Npffr2^{-/-}$ mice. Female $Npffr2^{-/-}$ mice showed comparable body weight and weight

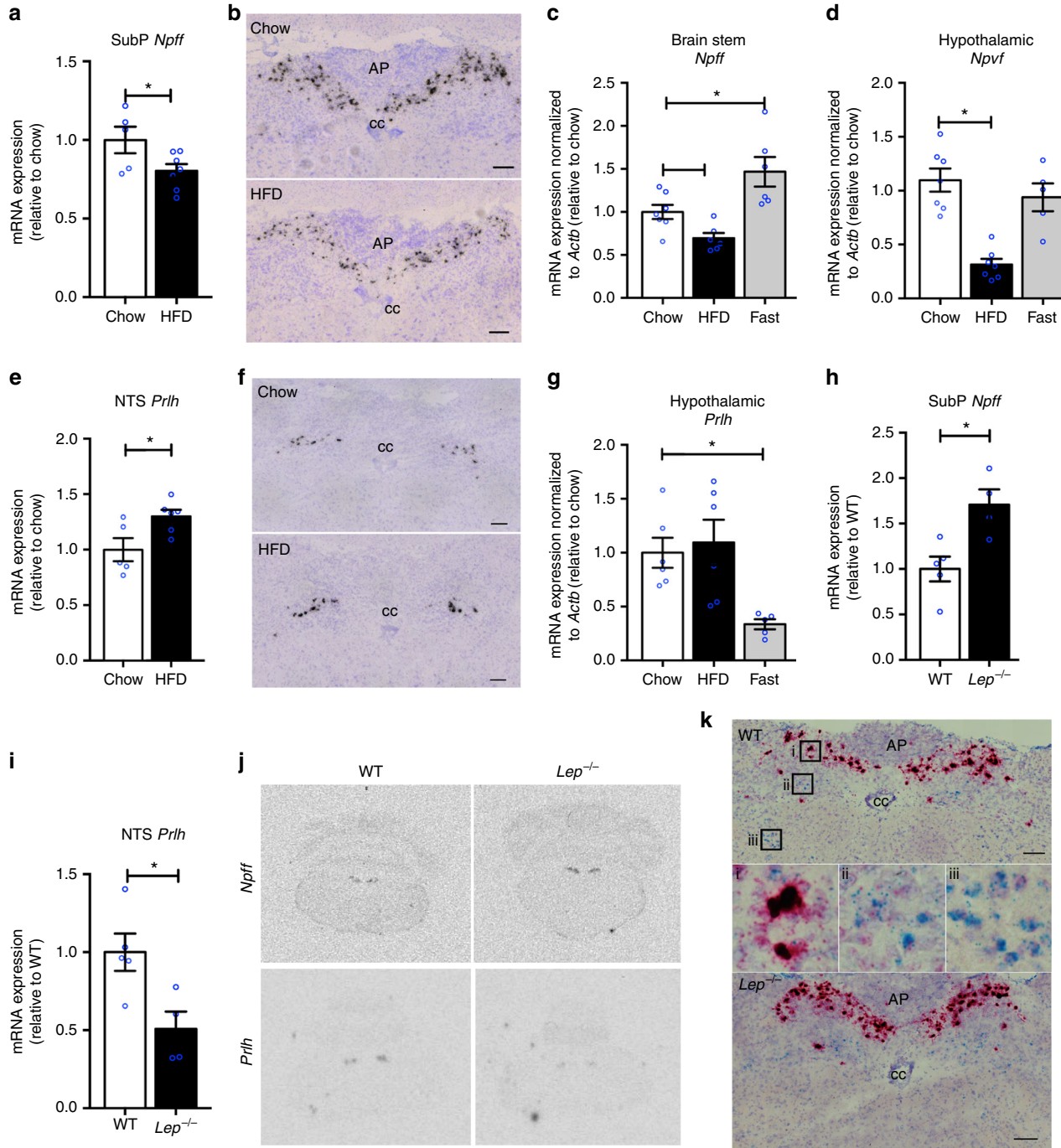

**Fig. 1** Expression of RFamide peptides in mouse brain under different nutrient status. **a** Expression of *Npff* mRNA in the subpostrema area (SubP) determined by in situ hybridization (ISH) under chow ($n = 5$) or HFD ($n = 7$) conditions. **b** Representative images of **a**. Scale bar = 100 μm. **c** Hindbrain expression of *Npff* determined by qPCR under chow ($n = 7$), HFD ($n = 6$), or 24 h fasting ($n = 6$) conditions. **d** Hypothalamic expression of *Npvf* determined by qPCR under chow ($n = 7$), HFD ($n = 7$), or 24 h fasting ($n = 5$) conditions. **e** Expression of PrRP (*Prlh*) in the nucleus tractus solitaries (NTS) determined by ISH under chow ($n = 5$) or HFD ($n = 6$) conditions. **f** Representative images of **e**. Scale bar = 100 μm. **g** Hypothalamic expression of PrRP (*Prlh*) determined by qPCR under chow ($n = 6$), HFD ($n = 6$), or 24 h fasting ($n = 5$) conditions. **h** Expression of *Npff* in the SubP determined by ISH in wild-type (WT) ($n = 5$) and $Lep^{-/-}$ ($n = 4$) mice. **i** Expression of PrRP (*Prlh*) in the NTS determined by ISH in WT ($n = 5$) and $Lep^{-/-}$ ($n = 4$) mice. **j** Representative ISH autoradiography images for **h** and **i**. **k** Images of dual labelling for *Lepr* (blue) and *Npff* (red) in WT and $Lep^{-/-}$ brains determined by RNAscope; (i) enlarged picture at SubP, (ii and iii) enlarged pictures at NTS. Data are mean ± SEM. *$p < 0.05$ as indicated by bar determined by one-way ANOVA and Tukey's HSD post-hoc test

gain to WT mice (Fig. 2e, f), but displayed a lean phenotype with significantly reduced individual and summed fat depots (Fig. 2g, h) (Supplementary Fig. 3c, d). Food intake in $Npffr2^{-/-}$ mice showed a significant reduction in males but not in females (Fig. 2i, j), suggesting that reduced feeding may contribute

to the reduced body weight gain in the male $Npffr2^{-/-}$ mice (Fig. 2b).

When fed on a HFD, $Npffr2^{-/-}$ mice of both genders showed significantly greater gains in body weight (Fig. 2k, l) and adiposity (Fig. 2c, d, g, h). Serum leptin levels were significantly elevated in

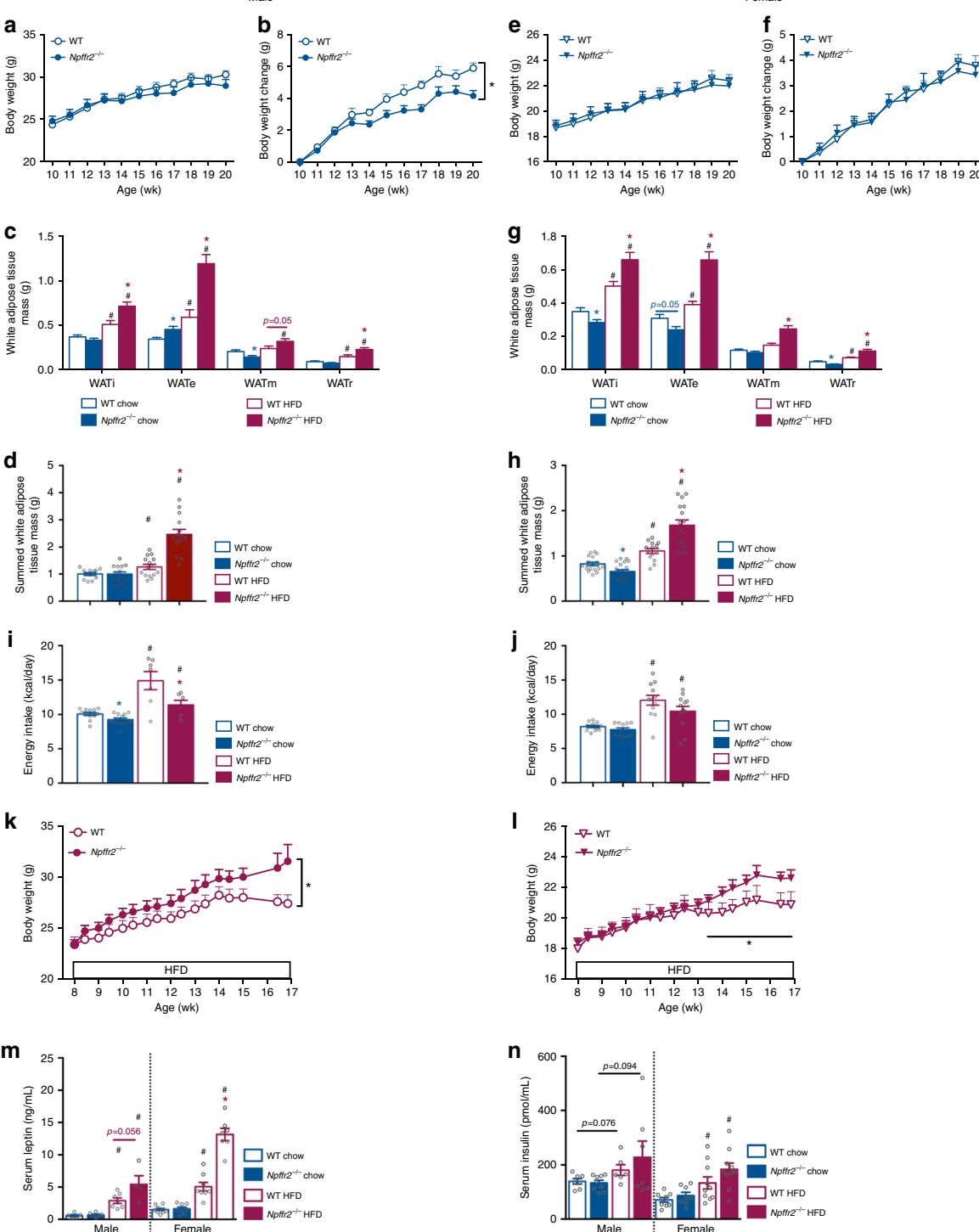

**Fig. 2** Exacerbated diet-induced obesity in *Npffr2*−/− mice. **a**, **b**, **e**, **f** Growth curve and body weight gain of mice on chow. Male: WT (open circle) *n* = 12, *Npffr2*−/− (filled circle) *n* = 8. Female: WT *n* = 11, *Npffr2*−/− *n* = 9. **c**, **g** Weights of individually dissected white adipose tissue depots of mice on chow or HFD. WATi, WATe, WATm and WATr represent inguinal, epididymal, mesenteric and retroperitoneal white adipose tissue, respectively. Male chow: WT *n* = 12, *Npffr2*−/− *n* = 15. Male HFD: WT *n* = 15, *Npffr2*−/− *n* = 14. Female chow: WT *n* = 17, *Npffr2*−/− *n* = 20. Female HFD: WT *n* = 14, *Npffr2*−/− *n* = 16. **d**, **h** Weights of summed white adipose tissue depots shown in **c**, **g**, respectively. **i**, **j** Energy intake in mice on chow or HFD. Male chow: WT *n* = 12, *Npffr2*−/− *n* = 13; male HFD: WT *n* = 7, *Npffr2*−/− *n* = 6. Female *n* = 12 per group. **k**, **l** Body weight curve on HFD. Male WT *n* = 13, *Npffr2*−/− *n* = 14. Female: WT *n* = 8, *Npffr2*−/− *n* = 10. **m**, **n** Serum leptin and insulin levels. Leptin: male chow (WT *n* = 5, *Npffr2*−/− *n* = 6), HFD (WT *n* = 7, *Npffr2*−/− *n* = 4); female chow (WT *n* = 7, *Npffr2*−/− *n* = 8), HFD (WT *n* = 8, *Npffr2*−/− *n* = 7); insulin: male chow (WT *n* = 7, *Npffr2*−/− *n* = 9), HFD (WT *n* = 6, *Npffr2*−/− *n* = 7); female chow (WT *n* = 9, *Npffr2*−/− *n* = 8), HFD (WT *n* = 10, *Npffr2*−/− *n* = 10). Tissues and serum samples were collected at 20 weeks of age for the chow cohort and at 17 weeks of age for the HFD cohort after 9 weeks of HFD. Data are mean ± SEM. One-way ANOVA (**c**, **d**, **g**, **h**, **i**, **j**, **m**, **n**) or repeated measures (**a**, **b**, **e**, **f**, **k**, **l**) were used to determine genotype or diet effects among groups. *$p < 0.05$ vs. WT of the same diet. #$p < 0.05$ vs. chow group of the same genotype

both HFD-fed WT and $Npffr2^{-/-}$ mice vs. their chow-fed counterparts with a greater increase in $Npffr2^{-/-}$ mice (Fig. 2m). In addition, HFD feeding led to increased serum insulin levels in both WT and $Npffr2^{-/-}$ mice, significantly so in female mice (Fig. 2n). Mice on HFD had higher calorie intake than their gender- and genotype-matched chow-fed counterparts (Fig. 2i, j); however, caloric intake on HFD was lower in the $Npffr2^{-/-}$ than WT mice, significantly so in male mice (Fig. 2i, j). This reduced energy intake in $Npffr2^{-/-}$ mice on HFD is in contrast to their exacerbated obesity suggesting mechanisms other than feeding are critical for this phenotype to occur.

**Impaired diet-induced thermogenesis in $Npffr2^{-/-}$ mice.** Male $Npffr2^{-/-}$ mice on chow showed comparable energy expenditure (normalized to metabolically active tissue (MAT) weight[26,27]) and physical activity to male WT mice (Fig. 3a, c, i, k), whereas female $Npffr2^{-/-}$ mice displayed a significant increase in light-phase energy expenditure and overall physical activity (Fig. 3b, d, j, l). Respiratory exchange ratio (RER) was unaltered in $Npffr2^{-/-}$ compared with WT mice in both genders (Fig. 3m, n).

Interestingly, when fed a HFD for 7 weeks, $Npffr2^{-/-}$ mice of both sexes had significantly reduced energy expenditure compared with gender-matched WT mice (Fig. 3e, f, i, j), with female but not male $Npffr2^{-/-}$ mice also exhibiting a reduction in overall activity (Fig. 3g, h). When energy expenditure under chow and HFD conditions was compared, both $Npffr2^{-/-}$ and WT mice on HFD showed significantly higher energy expenditure (Fig. 3i, j), consistent with an elevation in diet-induced thermogenesis. However, this increase in energy expenditure on HFD was significantly diminished in both male and female $Npffr2^{-/-}$ compared with gender-matched WT mice (Fig. 3i, j). Physical activity was significantly increased in female HFD-fed WT mice, but this was not observed in female or male $Npffr2^{-/-}$ mice (Fig. 3k, l). RER was lower under HFD conditions compared with chow conditions for both WT and $Npffr2^{-/-}$ mice (Fig. 3m, n), indicating a greater use of lipids as oxidative fuel during HFD feeding. Moreover, under HFD, female but not male $Npffr2^{-/-}$ mice had significantly lower RER than gender-matched WT mice (Fig. 3m, n). However, as obesity itself is known to decrease RER by enhancing lipid supply[28], the decreased RER in $Npffr2^{-/-}$ mice on HFD may be secondary to the increased obesity in these mice.

Taken together, these results demonstrate that NPFFR2 signalling regulates energy balance under basal chow conditions in a gender-dependent manner. However, when energy homoeostasis is challenged by an obesogenic diet, NPFFR2 signalling has a key role in mediating diet-induced adaptive thermogenesis uniformly in both genders.

**Impaired diet-induced brown fat thermogenesis in $Npffr2^{-/-}$ mice.** As diet-induced adaptive thermogenesis primarily takes place in BAT[7,8], we investigated BAT thermogenesis by measuring temperatures at the BAT ($T_{BAT}$) and lumbar back ($T_{Back}$) regions of freely moving mice using infrared imaging. We also calculated the temperature difference between the BAT and lumbar back regions ($\Delta T_{BAT-Back}$) as an indicator of BAT thermogenesis, since during increased BAT thermogenesis $\Delta T_{BAT-Back}$ would be expected to increase due to a direct contribution of heat arising from the BAT depot. $T_{BAT}$ and $T_{Back}$ in group-housed male or female mice on either chow or HFD were measured at baseline (14 weeks of age) and 6 weeks later (20 weeks of age). As results were consistent between the two genders, for clarity only data from female mice are presented here and male data are presented in Supplementary Fig. 4c-g. At baseline, WT and $Npffr2^{-/-}$ mice had comparable $T_{BAT}$, $T_{Back}$ and $\Delta T_{BAT-Back}$ (Fig. 4a, b).

After 6 weeks of HFD, WT mice showed a significant increase in $T_{BAT}$ and a trend towards an increase in $T_{Back}$, which resulted in an increase in $\Delta T_{BAT-Back}$ ($p = 0.07$ HFD-fed vs. chow-fed WT mice by $t$-test) (Fig. 4c). These increases induced by HFD in WT mice were not seen in $Npffr2^{-/-}$ mice (Fig. 4d), indicating an impaired BAT thermogenic response to HFD in the absence of NPFFR2 signalling. Of note, $T_{BAT}$ and $T_{Back}$ in chow-fed $Npffr2^{-/-}$ mice at 20 weeks of age were higher compared with WT mice (Supplementary Fig. 4a), which could be due to a differential response between $Npffr2^{-/-}$ and WT mice to handling stress. Importantly however, $\Delta T_{BAT-Back}$ was not significantly different between chow-fed $Npffr2^{-/-}$ and WT mice either at baseline or at the 6 weeks later time point (Supplementary Fig. 4b), suggesting comparable BAT thermogenesis between WT and $Npffr2^{-/-}$ mice under chow conditions. Male $Npffr2^{-/-}$ mice exhibited a similar thermogenic profile (Supplementary Fig 4c) and most importantly also exhibited the lack of increase in $\Delta T_{BAT-Back}$ in response to HFD (Supplementary Fig 4f,g) consistent with a gender-independent mechanism to control diet-induced thermogenesis in the absence of NPFFR2 signalling.

We examined BAT protein levels for UCP-1 and peroxisome proliferator-activated receptor-γ coactivator (PGC)-1α, the two key molecules involved in BAT adaptive thermogenesis[7,8,29]. BAT weight was significantly higher in $Npffr2^{-/-}$ mice (Fig. 4e) in association with a trend towards increased protein content in the BAT tissue (Fig. 4f). Under HFD, WT and $Npffr2^{-/-}$ mice significantly increased their BAT weights (Fig. 4e) that was attributable at least in part to an increase in protein content (Fig. 4f). The basal expression of BAT UCP-1 and PGC-1α per mg protein was unaltered in $Npffr2^{-/-}$ mice (Fig. 4g–i). When fed on HFD, WT but not $Npffr2^{-/-}$ mice showed marked increases in UCP-1 and PGC-1α levels per mg protein (Fig. 4g–i). When increases in BAT mass under HFD were taken into account, UCP-1 expression per BAT depot showed a significant increase in WT and $Npffr2^{-/-}$ mice on HFD; however, this increase was significantly smaller in $Npffr2^{-/-}$ compared with WT mice (Fig. 4j). Similarly, the PGC-1α expression per BAT depot was significantly increased in HFD-fed WT but not in $Npffr2^{-/-}$ mice (Fig. 4k).

To investigate whether cold-induced BAT thermogenesis is affected in $Npffr2^{-/-}$ mice we used single housing as a mild cold challenge compared with group housing at standard (22 °C) temperature[30]. Singly housed WT mice showed significantly higher $T_{BAT}$ but a comparable $T_{Back}$ to group-housed animals, which led to a significantly higher $\Delta T_{BAT-Back}$ in the single-housed mice (Fig. 4l). $Npffr2^{-/-}$ mice exhibited similar thermogenic responses to single housing as those seen in WT mice (Fig. 4m), indicating normal BAT thermogenic function in response to mild cold stress in the absence of NPFFR2 signalling. We further examined BAT thermogenesis at thermoneutrality (TN, 28 °C), using room temperature (RT) as the cold challenge. The $\Delta T_{BAT-Back}$ was significantly higher under RT compared with that under TN in both WT and $Npffr2^{-/-}$ mice, and this increase in $\Delta T_{BAT-Back}$ was similar in magnitude between WT and $Npffr2^{-/-}$ mice (Fig. 4n). Together, these results demonstrate that cold-induced thermogenesis is not controlled by NPFFR2 signalling.

We next examined BAT thermogenesis in WT and $Npffr2^{-/-}$ mice fed a HFD under TN condition. $Npffr2^{-/-}$ mice fed a HFD under TN exhibited significantly greater weight and fat gain compared to WT mice (Fig. 4o–q). Importantly, WT mice on HFD showed significant increases in $T_{BAT}$ and $\Delta T_{BAT-Back}$ compared with chow-fed animals (Fig. 4r), whereas no such increases were seen in $Npffr2^{-/-}$ mice (Fig. 4s).

As leptin has been shown to have a role in diet-induced BAT thermogenesis[31], we investigated the influence of leptin action in the absence of NPFFR2 signalling by examining pSTAT3

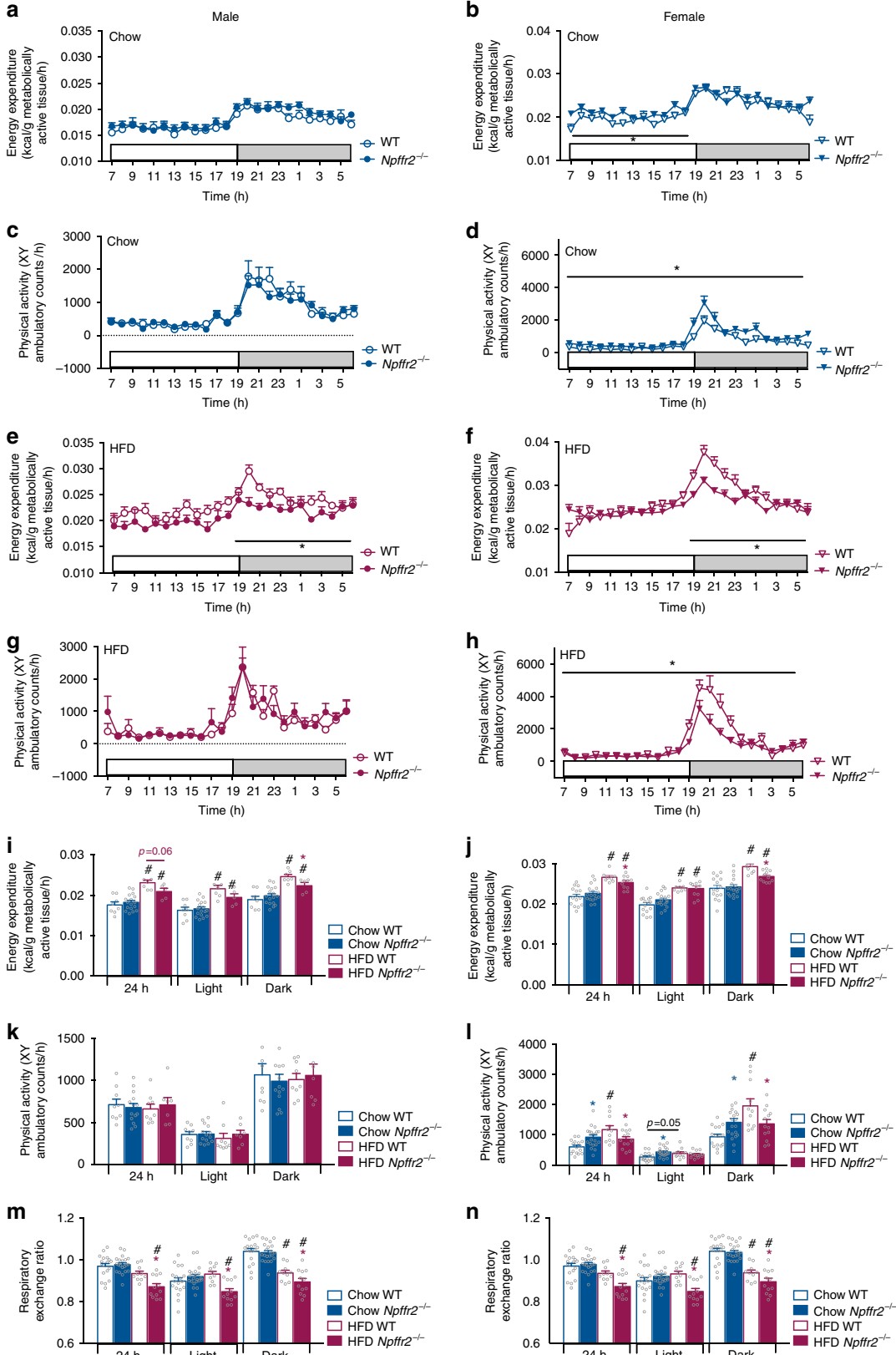

**Fig. 3** Energy homoeostasis parameters in WT and $Npffr2^{-/-}$ mice under chow and HFD conditions. **a–d** Time-course of energy expenditure (**a**, **b**) and physical activity (**c**, **d**) in WT (open circle) and $Npffr2^{-/-}$ (filled circle) mice of both genders on chow. **e–h** Time-course of energy expenditure (**e**, **f**) and physical activity (**g**, **h**) in WT and $Npffr2^{-/-}$ mice of both genders on HFD. **i–n** Energy expenditure (**i**, **j**), physical activity (**k**, **l**) and respiratory exchange ratio (**m**, **n**) averaged over 24 h in WT and $Npffr2^{-/-}$ mice fed on chow or HFD. Male chow: WT $n = 8$, $Npffr2^{-/-}$ $n = 14$; male HFD: WT $n = 9$, $Npffr2^{-/-}$ $n = 7$. Female chow: WT $n = 14$, $Npffr2^{-/-}$ $n = 17$; female HFD: WT $n = 8$, $Npffr2^{-/-}$ $n = 11$. Data are mean ± SEM. One-way ANOVA (**i–n**) or repeated measures (**a–h**) were used to determine genotype or diet effects among groups. *$p < 0.05$ vs. WT of the same diet. #$p < 0.05$ vs. chow group of the same genotype

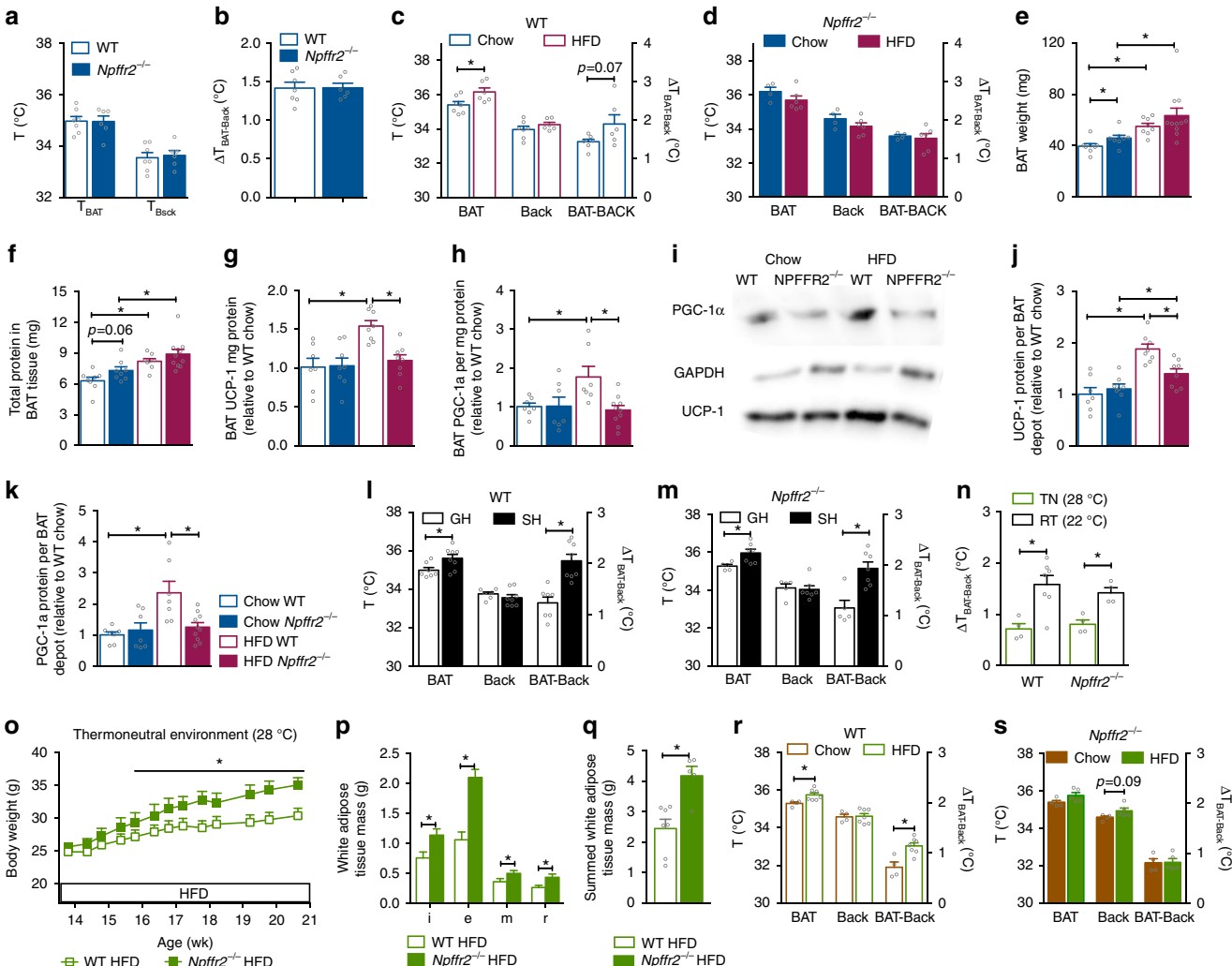

**Fig. 4** Impaired diet-induced but not cold exposure-induced brown adipose tissue thermogenesis in $Npffr2^{-/-}$ mice. **a, b** Skin temperature at lumbar back ($T_{Back}$) and brown adipose tissue ($T_{BAT}$) by infrared imaging in chow-fed WT ($n = 7$) and $Npffr2^{-/-}$ ($n = 6$) mice at 14 weeks of age housed at room temperature (RT 22 °C). **b** Temperature difference ($\Delta T_{BAT-Back}$) between $T_{BAT}$ and $T_{Back}$ in chow-fed WT ($n = 7$) and $Npffr2^{-/-}$ ($n = 6$) mice. **c, d** Effect of 6-weeks HFD on $T_{BAT}$, $T_{Back}$ and $\Delta T_{BAT-Back}$ in WT (chow $n = 7$, HFD $n = 6$) and $Npffr2^{-/-}$ (chow $n = 4$, HFD $n = 6$) mice. **e, f** BAT weight and protein content in mice on chow (WT $n = 8$, $Npffr2^{-/-}$ $n = 8$) or 6 weeks of HFD (WT $n = 8$, $Npffr2^{-/-}$ $n = 11$) at 20 weeks of age. **g, h** BAT UCP-1 and PGC-1α protein expression per mg protein. $N$ values as in **e, f**. **i** Representative western blotting image of **g, h**. **j, k** BAT UCP-1 and PGC-1α protein expression per BAT depot. $N$ values as in **e, f**. **l, m** Effect of 3 weeks single housing (SH) as mild cold exposure vs. group housing (GH) on $T_{BAT}$, $T_{Back}$ and $\Delta T_{BAT-Back}$ in WT (GH $n = 7$, SH $n = 8$) and $Npffr2^{-/-}$ (GH $n = 5$, SH $n = 7$) mice. **n** Effect of RT (22 °C) as mild cold exposure vs. thermoneutrality (28 °C, TN) on $T_{BAT}$, $T_{Back}$ and $\Delta T_{BAT-Back}$ in WT (RT $n = 7$, TN $n = 4$) and $Npffr2^{-/-}$ (RT $n = 4$, TN $n = 4$) mice. **o** Body weight of WT ($n = 7$) and $Npffr2^{-/-}$ ($n = 5$) mice fed on HFD for 7 weeks at TN. **p, q** Weights of white adipose tissue depots of WT ($n = 7$) and $Npffr2^{-/-}$ ($n = 5$) mice fed on HFD for 7 weeks at TN; i, e, m and r stand for inguinal, epididymal, mesenteric and retroperitoneal white adipose tissue, respectively. **r, s** Effect of 7 weeks of HFD on $T_{BAT}$, $T_{Back}$ and $\Delta T_{BAT-Back}$ in WT (Chow $n = 4$, HFD $n = 7$) and $Npffr2^{-/-}$ (Chow $n = 4$, HFD $n = 5$) mice at TN. Data are mean ± SEM. One-way ANOVA or repeated measures (**o**) were used to determine genotype, diet or treatment effects among groups. **\*\***$p < 0.05$ as indicated by black bar

expression using immunohistochemistry in WT and $Npffr2^{-/-}$ mice in response to leptin stimulation. Chow-fed WT and $Npffr2^{-/-}$ mice had comparable basal pSTAT3 levels in the Arc of the hypothalamus and showed a more than twofold increase in pSTAT3 activation in response to leptin—administered either intraperitoneal (i.p.) (2 μg/g body weight) or intracerebroventricular (i.c.v.) (1.25 μg)—(Fig. 5a, b). WT and $Npffr2^{-/-}$ fed on HFD showed similar pSTAT3 in the Arc at baseline (Fig. 5a, b) and marked increases in pSTAT3 activation in response to i.p. leptin which was significantly greater in $Npffr2^{-/-}$ mice (Fig. 5a, b). These results demonstrate that leptin-induced pSTAT3 activation is not impaired in $Npffr2^{-/-}$ mice, suggesting that defective leptin signalling is unlikely to contribute to the impaired

BAT thermogenesis and exacerbated diet-induced obesity seen in $Npffr2^{-/-}$ mice.

## NPFFR2 signalling controls BAT function via an Arc NPY circuit.
Consistent with a predominantly central action of NPFFR2, the $Npffr2$ mRNA is highly expressed in the hypothalamus with only minute levels in subcutaneous white adipose tissue and not detectable in BAT (Fig. 5c). Moreover, hypothalamic $Npffr2$ mRNA expression was significantly decreased in HFD versus Chow-fed mice (Fig. 5d), demonstrating the responsiveness of the hypothalamic NPFFR2 signalling to changes in energy status. ISH using RNAscope identified the most

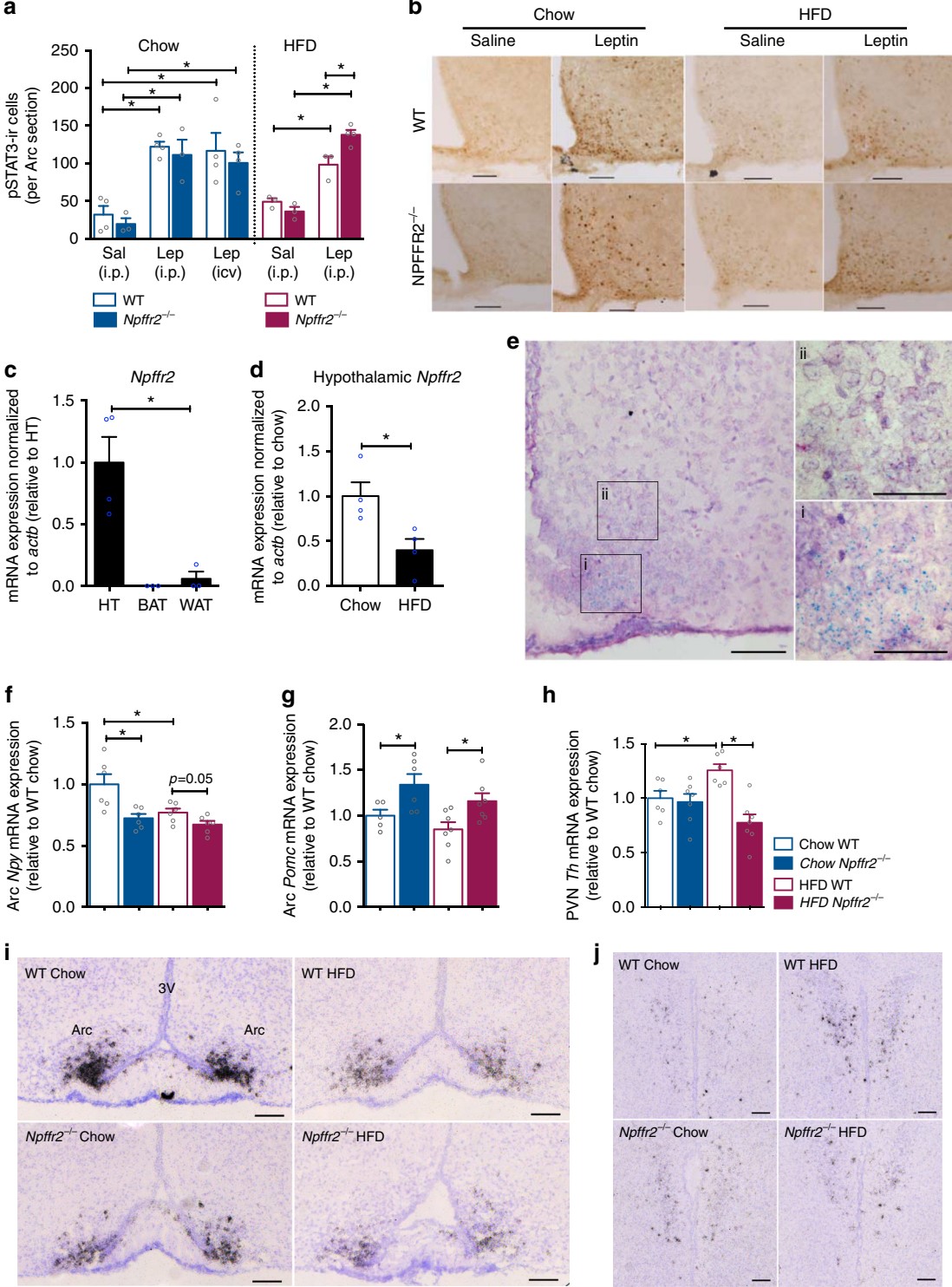

**Fig. 5** NPFFR2 signalling regulates the Arc NPY – PVN-TH pathway. **a** pSTAT3-ir cells in the Arc of male WT and *Npffr2*⁻/⁻ mice fed on either chow or 6 weeks of HFD. Brains were collected at 30 min after an i.p. injection of saline or leptin (2 µg/g body weight). Another cohort of these mice received intracerebroventricular (i.c.v.) injection of saline or leptin (1.25 µg) at 30 min prior to perfusion and brain collection. *n* = 3–4 brains per group. Scale bar = 100 µm. **b** Representative images of IHC on pSTAT3 used for pSTAT3-ir cell quantification after i.p. injection of either saline or leptin shown in **a**. **c**, **d** qPCR analysis on the mRNA expression of *Npffr2* in hypothalamus (HT), brown adipose tissue (BAT), subcutaneous white adipose tissue (WAT), hypothalamic expression on chow and 9 weeks of HFD. *n* = 3–4 per group. **e** Image of chromogenic ISH for *Npffr2* expression at the Arc. Scale bar = 100 µm in the image on the left, 50 µm in images (i) and (ii). Representative images from 3 independent experiments. **f**, **g**, **h** The expression of *Npy* and *Pomc* by ISH at Arc, and *Th* at PVN in WT and *Npffr2*⁻/⁻ mice fed on chow or 9 weeks of HFD. *n* = 5–7 per group. **i** Representative images of ISH on *Npy* expression at the Arc quantified in **f**. Scale bar = 100 µm. **j** Representative images of ISH on *Th* expression at the PVN quantified in **h**. Scale bar = 100 µm. Data are mean ± SEM. One-way ANOVA was used to determine the treatment (**a**), tissue type (**c**) diet and genotype (**d**, **b**, **g**, **h**) effects among groups. **$p < 0.05$ as indicated by bar. Arc, arcuate nucleus. PVN, paraventricular nucleus of the hypothalamus

abundant expression of *Npffr2* in the Arc (Fig. 5e) with high concentration at the ventral medial region (Fig. 5e–(i)) and little expression at the dorsal lateral part (Fig. 5e-(ii)). The accessibility of the Arc by RFamide peptides is supported by studies showing direct projections and actions of RFamide peptide-expressing neurons (e.g., *Npvf*- and *Kiss1*-expressing neurons) towards the Arc[20,32,33] and hormone-like action of NPFF via secretion into the blood stream and cerebrospinal fluid[34,35]. Using retrograde tracing we show that projections from the SubP, where NPFF is predominantly expressed, exist that connect to the Arc (Supplementary Fig. 5a–e). Thus, red fluorescent beads were injected into the Arc and the fluorescent signal was examined 2 weeks later in the brain stem. Strong red fluorescence was seen at the injection site, which had some 'crossover' to the green channel due to the brightness of the tracer[36]. Importantly, specific red fluorescence-labelled cells and fibres were observed in the area postrema (AP) and SubP without any crossover to the green channel (Supplementary Fig. 5c–e). Specific red fluorescence-labelled cells and fibres were also seen in the NTS (Supplementary Fig. 5f–h), although appear to be fewer than those in the AP and SubP (Supplementary Fig. 5e).

To examine whether a lack of NPFFR2 alters the expression of related receptors of cross-reacting ligands, we determined the hypothalamic expression of *Kiss1r* and *Npffr1*—the cognate receptor for Kisspeptin and NPVF, respectively[20,32,33,37] — in $Npffr2^{-/-}$ and WT mice. Although the expression of *Npffr1* was not detectable in either WT or $Npffr2^{-/-}$ hypothalamic tissue (data not shown), the hypothalamic expression of *Kiss1r* was not significantly different between WT and $Npffr2^{-/-}$ mice (Supplementary Fig. 6a) and was not altered by HFD (Supplementary Fig. 6b), arguing against a compensatory action from related receptor signalling due to NPFFR2 deletion and contrasting the responsiveness of hypothalamic *Npffr2* to the changed energy status (Fig. 5d).

As NPY signalling pathways originating from the Arc are known to have a key role in the regulation of energy homoeostasis[38], we investigated whether its expression is influenced by lack of NPFFR2 signalling. Interestingly, $Npffr2^{-/-}$ mice under chow conditions exhibited significantly lower *Npy* mRNA levels in the Arc compared with WT mice (Fig. 5f, i). Six weeks of HFD feeding resulted in an approximately 30% reduction in Arc *Npy* mRNA expression in WT but not in $Npffr2^{-/-}$ mice compared with their chow counterparts (Fig. 5f, i). In contrast, expression of pro-opiomelanocortin (*Pomc*) mRNA was significantly increased in $Npffr2^{-/-}$ compared with WT mice under both chow and HFD conditions without significant effect of diet in either genotype (Fig. 5g).

To test whether NPFFR2 signalling uses a known NPY-dependent relay via tyrosine hydroxylase (TH)-positive neurons in the paraventricular nucleus (PVN)[38], we investigated the expression levels of PVN-*Th* mRNA by ISH. Under chow PVN-*Th* mRNA was unaltered in $Npffr2^{-/-}$ mice (Fig. 5h, j); however, when fed a HFD for 6 weeks, PVN-*Th* mRNA levels were significantly increased in WT but not in $Npffr2^{-/-}$ mice (Fig. 5h, j). Thyrotropin-releasing hormone (*Trh*) mRNA showed a strong trend towards increasing in WT but not $Npffr2^{-/-}$ mice in response to HFD (Supplementary Fig. 6c). *Oxt* mRNA levels were significantly reduced in response to HFD in both WT and $Npffr2^{-/-}$ mice without significant differences between genotypes (Supplementary Fig. 6d). Together, these results indicate that the responsiveness of PVN-TH neurons to a HFD in terms of *Th* expression level is dependent on NPFFR2 signalling.

**NPFFR2 signalling regulates bone metabolism**. As Arc NPY neurons are also involved in the regulation of bone homoeostasis[39], we used micro-computed tomography (μCT) analysis on isolated femora. Male $Npffr2^{-/-}$ mice, while having slightly reduced femoral length (Fig. 6a), have a significant increase in cancellous bone mass (bone volume over total volume, BV/TV) compared with WT mice (Fig. 6b), associated with a significant increase in trabecular thickness and trabecular number (Fig. 6c, d). Cortical bone volume and cortical thickness also showed a trend towards increasing in male $Npffr2^{-/-}$ mice (Fig. 6i,j). In contrast, female $Npffr2^{-/-}$ mice had unaltered femoral length and significant increases in cortical volume and cortical thickness compared with WT mice (Fig. 6k, l) without significant changes in cancellous bone parameters including cancellous bone volume (Fig. 6f), trabecular thickness (Fig. 6g) and trabecular number (Fig. 6h).

**NPFFR2 signalling directly acts on Arc NPY neurons**. To investigate whether NPFFR2 signalling directly regulates NPY-expressing neurons, a translating ribosome affinity purification (TRAP) approach was employed. Thus, GFP-L10a fusion protein encoding Rosa26CAG-EGFP (TRAP) mice[40] were crossed onto our $Npy^{cre/+}$ mice to generate the $Npy^{cre/+}$; TRAP$^{lox/lox}$ mice (NPY-TRAP), where the ribosomal L10a-GFP fusion protein is only produced in NPY neurons (Fig. 7a). As a control we used $Ins^{Cre/+}$; TRAP$^{lox/lox}$ mice (Ins-TRAP) in which the L10a-GFP fusion protein is expressed ubiquitously in the hypothalamus, as well as TRAP$^{lox/lox}$ (WT-TRAP) mice without any *cre* (Supplementary Fig. 7a). Quantitative PCR (qPCR) performed on mRNA isolated by immunoprecipitation with an anti-green fluorescent protein (GFP) antibody against the GFP-tagged ribosomal unit confirmed a significant enrichment of GFP and *Npy* transcripts in the immunoprecipitated RNA relative to the corresponding unprocessed RNA (Input) sample (Supplementary Fig. 7b) with no *Npy* enrichment in mice without the activation of TRAP (WT-TRAP) (Fig. 7b). Importantly, *Npffr2* mRNA was also highly enriched indicating that *Npffr2* is expressed on the NPY neurons in the Arc (Fig. 7c). Interestingly, *Npffr2* enrichment was not observed in a NPY neuronal population from the amygdala where *Npffr2* mRNA was actually deriched (Supplementary Fig. 7c).

Consistent with the TRAP results, dual ISH using RNAscope shows that a subset of NPY neurons in the Arc express *Npffr2* (Fig. 7d-(i))—results also confirmed by recent studies employing a single-cell sequencing approach on Arc neurons[41,42]. Of note, there are Arc NPY neurons that do not express *Npffr2* (Fig. 7d-(ii)), as well as non-NPY neurons that do express *Npffr2* (Fig. 7d-(iii)), suggesting that NPFFR2 signalling could also affect other Arc neurons, allowing for the possibility of an indirect action of NPFFR2 signalling on NPY neurons. A similar distribution of *Npffr2* expression with an overlap with NPY neurons can also be found in the DMH (dorsomedial hypothalamic nucleus; Supplementary Fig. 7e), a hypothalamic nuclei also known to have influences on BAT function[43].

It is worth noting that although the neurochemical identities of these Arc *Npffr2*-expressing non-NPY neurons remains to be determined, they are unlikely to be the *Pomc*-expressing neurons, as little colocalization of *Pomc* and *Npffr2* mRNA was observed (Supplementary Fig 7e), consistent with single-cell sequencing findings[41].

As NPFFR2 has been shown to couple to the activation of the class of G-proteins that inhibit cAMP accumulation[44,45], we examined the effect of NPFF-induced NPFFR2 signalling on the phosphorylation of the cAMP-responsive element binding protein (CREB) — a downstream responder to forskoline-induced increases in cAMP levels — using NPY-GFP mice[46,47]. I.c.v. injection of forskoline produced a marked increase in pCREB immunoreactivity in approximately 60% of Arc NPY

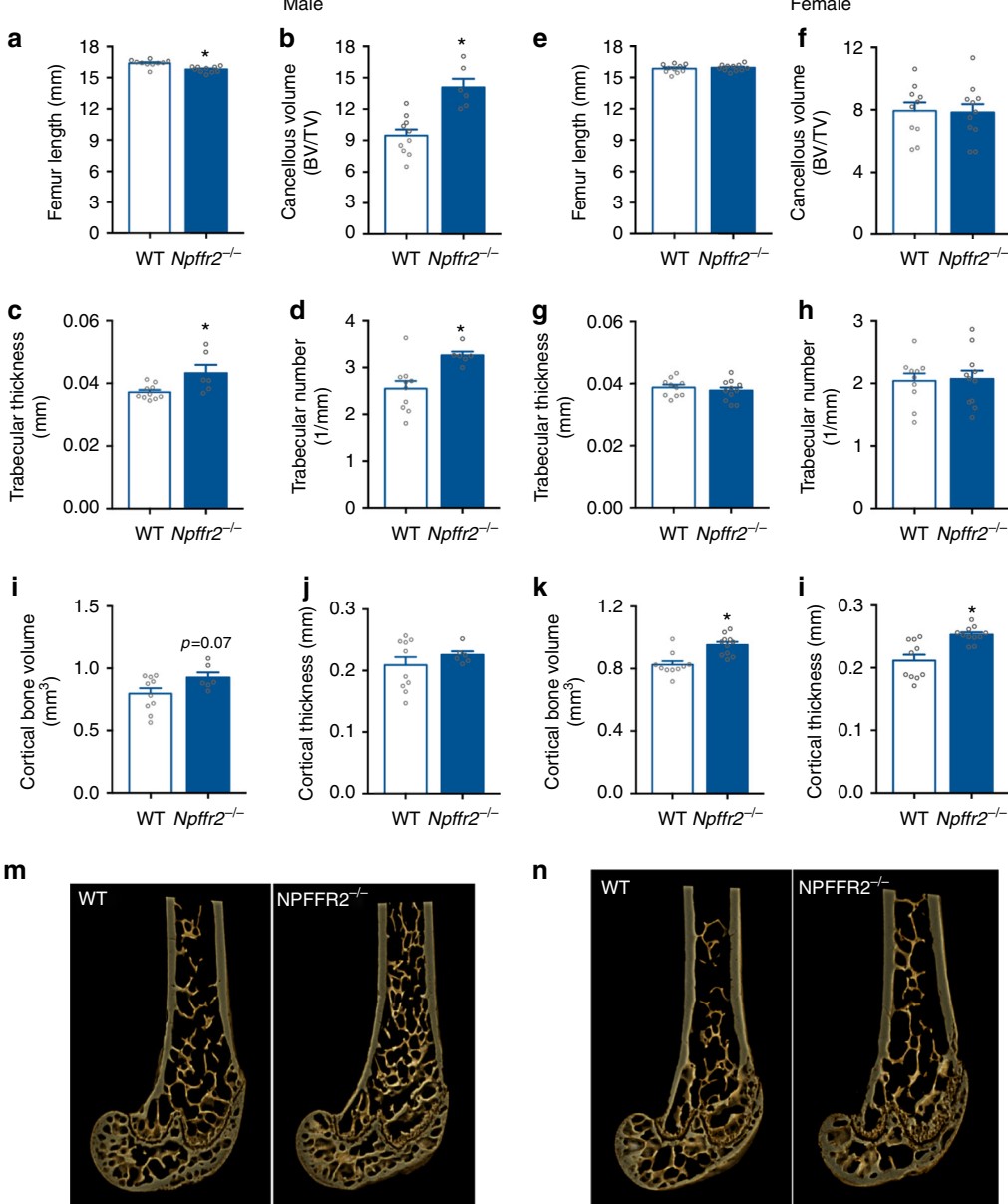

**Fig. 6** NPFFR2 signalling regulates bone homoeostasis. **a**, **e** Femoral length of male and female WT and $Npffr2^{-/-}$ mice. **b–d**, **f–h** Cancellous bone volume, trabecular thickness and trabecular number of the distal femur in male and female WT and $Npffr2^{-/-}$ mice. **i**, **j**, **k**, **l** Cortical bone volume and cortical thickness of the distal femur in male and female WT and NPFFR2$^{-/-}$ mice. **m**, **n** Representative photomicrographs of the distal femoral metaphysis of male and female WT and $Npffr2^{-/-}$ mice. Male: WT $n = 10$, $Npffr2^{-/-}$ $n = 6$. female: WT $n = 10$, $Npffr2^{-/-}$ $n = 11$. BV/TV, bone volume over total volume. Data were from mice at 19–20 weeks of age. Data are mean ± SEM. One-way ANOVA was used to determine difference among groups. **\*\***$p < 0.05$ vs. WT

neurons compared with saline-injected controls (Fig. 7e). Although NPFF alone had no apparent effect on cAMP/CREB activation (Fig. 7e), co-administration of NPFF with forskolin significantly reduced the number of Arc NPY neurons colocalized with pCREB-ir (approximately 20%) (Fig. 7f, Supplementary Table 2), demonstrating a direct action of NPFFR2 signalling on NPY neurons in the Arc.

**NPFFR2 signalling on NPY neurons regulates diet-induced thermogenesis.** The direct role of NPFFR2 signalling on NPY neurons in the control of energy homoeostasis in vivo was tested by an inducible conditional NPFFR2 knockout (KO) mouse model ($Npy^{creER2/+},Npffr2^{lox/lox}$) where $Npffr2$ deletion occurs specifically in NPY neurons upon induction by tamoxifen. As controls, $Npy^{creER2/+},Npffr2^{lox/lox}$ mice without tamoxifen

treatment and $Npy^{creER2/+},Npffr2^{lox/+}$ mice treated with tamoxifen were used. Tamoxifen-induced NPY neuron-specific $Npffr2$ gene deletion was confirmed by PCR in NPY-expressing tissues including the hypothalamus, olfactory bulb and adrenal gland, but was absent in tissues with no NPY expression (e.g., the liver) or in tissues of mice that were not treated with tamoxifen (Supplementary Fig. 8).

Conditional $Npy^{creER2/+},Npffr2^{lox/lox}$ KO (referred as 'conditional KO' hereafter) mice and control mice on chow diet were not significantly different with regard to body weight (Fig. 8a) or adiposity (Fig. 8b, c). Five weeks of HFD led to the development of obesity in both conditional KO and control mice with a trend of higher body weight (Fig. 8d) and significant greater increases in fat mass in the conditional KO mice (Fig. 8b, c), consistent with the results from the global $Npffr2^{-/-}$ mice.

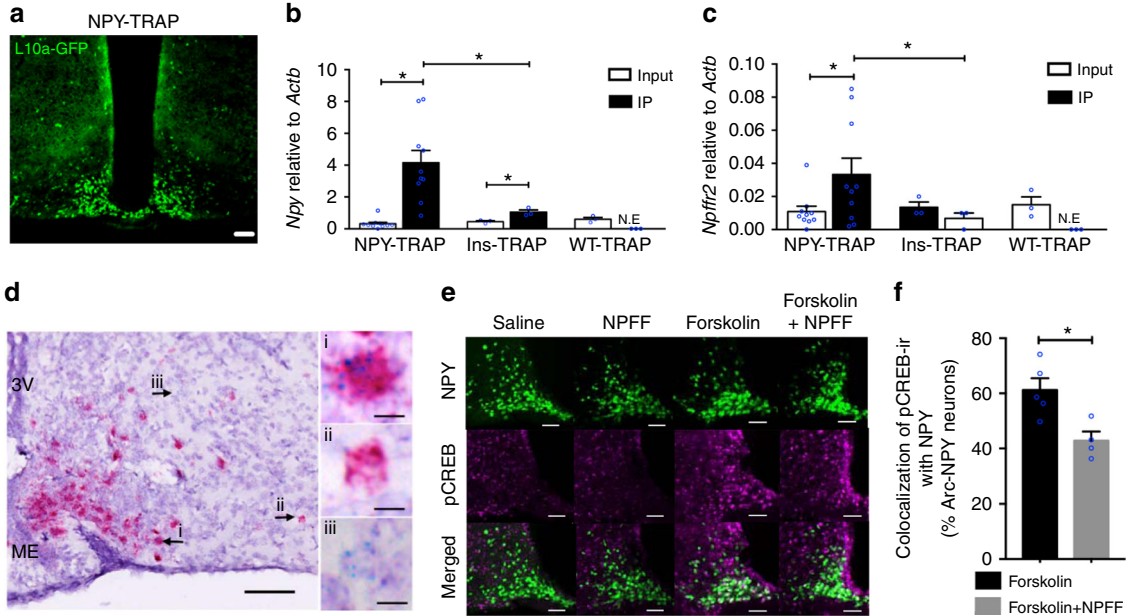

**Fig. 7** Colocalization and effects of direct NPFFR2 signalling on NPY neurons. **a** Representative image of GFP expression in the Arc of a NPY-TRAP mouse brain. Scale bar = 100 μm. **b, c** Quantification of the expression of *Npy* and *Npffr2* mRNA in the input and immunoprecipitated (IP) RNA isolated from the Arc of NPY-TRAP (*n* = 10), Ins-TRAP (*n* = 3) and WT-TRAP (*n* = 3) mice. One-way ANOVA was used to determine difference between groups. **$p < 0.05$ as indicated by bar. N.E, gene is not expressed. **d** Double chromogenic ISH for *Npy* (red) and *Npffr2* (blue) expression in the Arc. Scale bar = 100 μm in the image on the left, 10 μm in images (i), (ii) and (iii). Representative image from three independent experiments. **e** Representative images of IHC of brain sections for pCREB (magenta) in NPY-GFP mice that received i.c.v. either saline, NPFF (150 μM), Forskolin (2.5 mM) or NPFF (150 μM) + Forsoline (2.5 mM). Brains were collected 30 min after i.c.v. injection. Scale bar = 100 μm. **f** Quantification of pCREB-ir-positive neurons colocalized with Arc GFP-labelled NPY neurons expressed as percentage of Arc NPY neurons. Forskolin *n* = 5, Forskolin + NPFF *n* = 4. Data are mean ± SEM. One-way ANOVA was used to determine difference between groups. **$p < 0.05$ as indicated by bars

Both control and conditional KO mice had greater energy intake on HFD than on chow with no significant difference between genotypes under either diet (Fig. 8e). Energy expenditure did not significantly differ between control and conditional KO mice under chow (Fig. 8f, l) or after 4 weeks of HFD (Fig. 8i, l); however, a strong trend towards a decrease in energy expenditure in the HFD conditional KO mice in the early period of the dark phase was observed (Fig. 8i). Physical activity was not significantly altered by genotype or diet (Fig. 8g, j, m). Interestingly, RER, although not significantly different between control and conditional KO mice under chow (Fig. 8h, n), was markedly higher in conditional KO mice when fed on HFD (Fig. 8k, n), suggesting a reduced reliance on lipids as oxidative fuel when there is an increased dietary lipid influx likely contributing to the greater fat gain in the conditional KO mice under HFD (Fig. 8b, c).

We also examined BAT thermogenesis in an additional cohort of mice at an earlier time point, i.e., 10 days after starting HFD. Both control and conditional KO mice showed significantly increased $T_{BAT}$ and an unaltered $T_{Back}$ (Fig. 9a, b), leading to a significant increase in $\Delta T_{BAT-Back}$ (Fig., 9c). The increase in $\Delta T_{BAT-Back}$ by HFD was significantly less in the conditional KO mice compared with controls (Fig. 9c), suggesting that diet-induced BAT activation is impaired in mice lacking NPFFR2 in NPY neurons in the early phase of HFD exposure. We further conducted indirect calorimetry during the transition period from chow to HFD where mice were monitored for the first 5 days of HFD feeding. Both control and conditional KO mice showed increased caloric intake once the diet was switched from chow to HFD (Fig. 9d). Importantly, although energy expenditure was increased on HFD compared with chow in both conditional KO and control mice, this increase was lower in the conditional KO mice with significant effects most evident during the light phase

(Fig. 9e, f). These data show a key role of NPFFR2 signalling in NPY neurons in the quick implementation of HFD diet-induced thermogenesis in BAT.

**NPFFR2 signalling on NPY neurons controls bone formation.** Conditional KO mice displayed an increase in femur length, femur bone mineral density and a trend towards an increase in femur bone mineral content (Fig. 9g–i), demonstrating that specific signalling of NPFFR2 in NPY neurons is important for the control of bone formation. Further μCT analysis revealed that the conditional KO mice had significant increases in both trabecular and cortical bone mass, i.e., significant increases in cancellous bone volume, trabecular thickness, trabecular number, cortical bone volume and cortical thickness (Fig. 9j–o). These results are consistent with the findings in the global *Npffr2*[-/-] mice and confirm that NPFFR2 signalling also has a major role in the central regulation of bone homoeostasis via direct actions on NPY neurons.

## Discussion

This study demonstrates that HFD-induced adaptive thermogenesis is dependent on functional signalling through NPFFR2 receptors. Loss of NPFFR2 results in significantly greater diet-induced increases in body weight and adiposity, predominantly due to a lack of BAT activation and a subsequent reduction in energy expenditure. One of the critical relay points for this thermogenic control is the Arc, where NPFFR2 signalling provides the tone for maintaining a baseline level of NPY expression necessary to exert homoeostatic control, via direct and possibly also indirect actions. Lack of this NPFFR2-induced tone on NPY neurons reduces the capacity of these neurons to control TH-

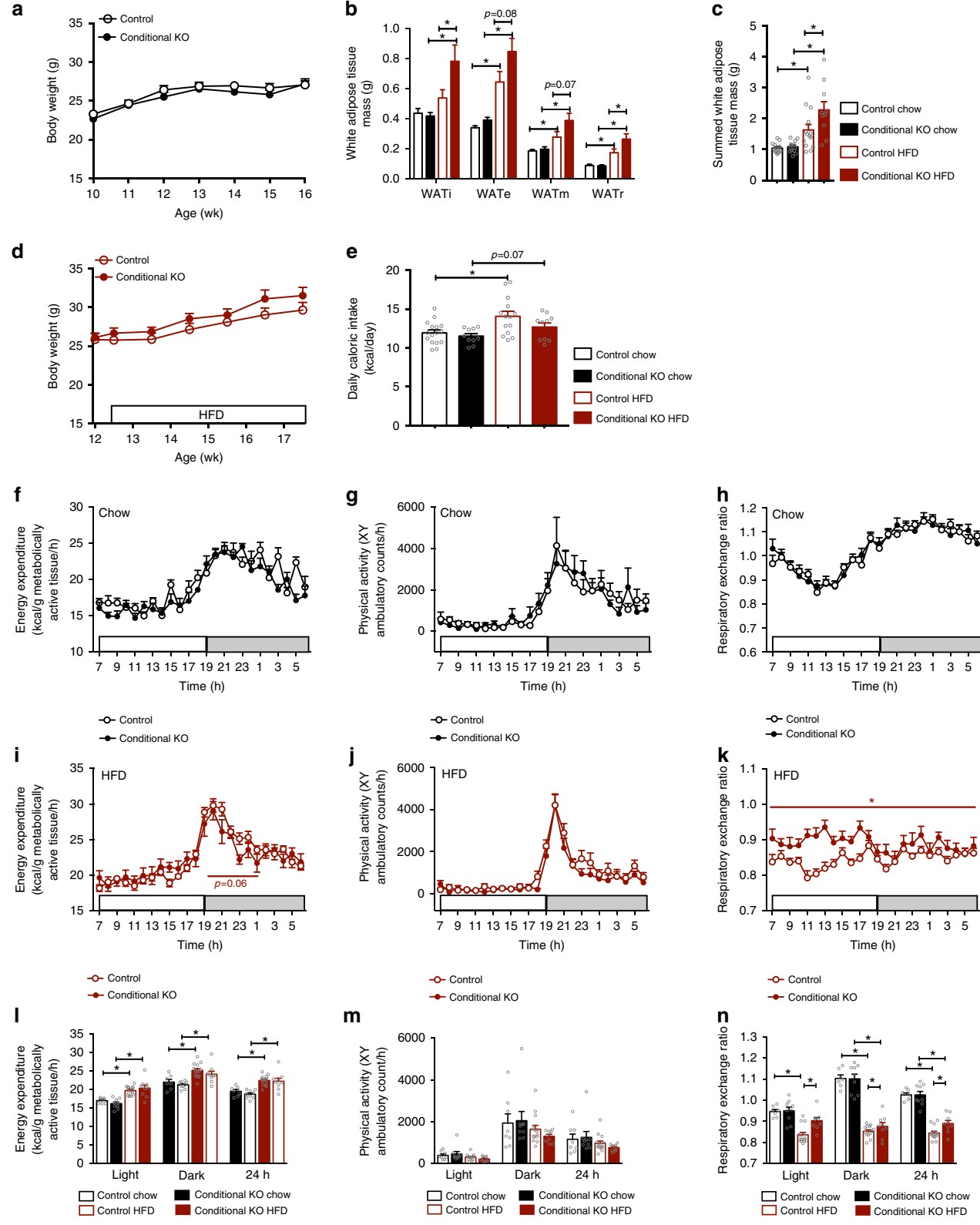

positive neurons in the PVN, with a consequent reduction in BAT function[38].

Our results show that NPFFR2 signalling regulates energy metabolism and bone mass in a gender-dependent manner under basal chow-fed conditions. However, under obesogenic conditions, NPFFR2 signalling exerts a uniform action in both genders to regulate diet-induced thermogenesis. It is interesting to note

that there is a gender difference in the response of physical activity to energy excess, in that female but not male mice increased their activity in response to HFD. This increase in physical activity during HFD in female mice is consistent with observations in humans suggesting that increased physical activity can be an adaptive response to overfeeding[48]. Our study further shows that this increase in physical activity associated

**Fig. 8** Altered diet-induced obesity and energy homoeostasis in mice with conditional NPFFR2 deletion in NPY neurons under chow and HFD conditions. **a**, **d** Growth curve of male conditional KO and control mice on chow and HFD. Chow: control $n = 14$, conditional KO $n = 13$. HFD: control $n = 14$, conditional KO $n = 10$. **b**, **c** Weights of individual and summed white adipose tissue depots in male conditional KO and control mice under chow and HFD conditions. WATi, WATe, WATm and WATr represent inguinal, epididymal, mesenteric and retroperitoneal white adipose tissue, respectively. *N* values are as in **a**, **d**. **e** Energy intake in male conditional KO and control mice under chow and HFD conditions. Chow: control $n = 16$, conditional KO $n = 10$. HFD: control $n = 14$, conditional KO $n = 10$. **f**, **i**, **l** Energy expenditure time-course and averages over the light, dark and 24 h periods in male conditional KO and control mice under chow and HFD conditions. Chow: control $n = 8$, conditional KO $n = 9$. HFD: control $n = 14$, conditional KO $n = 9$. **g**, **j**, **m** Physical activity time-course and averages over the light, dark and 24 h periods in male conditional KO and control mice under chow and HFD conditions. *N* values are as in **f**, **i**, **l**. **h**, **k**, **n** Respiratory exchange ratio time-course and averages over the light, dark and 24 h periods in male conditional KO and control mice under chow and HFD conditions. *N* values are as in **f**, **i**, **l**. Gene deletion was induced by tamoxifen in male $Npy^{creER2/+},Npffr2^{lox/lox}$ mice at 6 weeks of age (Conditional KO). Male $Npy^{creER2/+},Npffr2^{lox/lox}$ mice without tamoxifen and $Npy^{creER2/+},Npffr2^{lox/+}$ mice treated with tamoxifen were used as controls. Data are mean ± SEM. One-way ANOVA **b**, **c**, **e**, **l**, **m**, **n** or repeated measures **a**, **d**, **f–h**, **i–k** was used to determine difference among groups. **\*\****p* < 0.05 as indicated by bar

with HFD is absent in female $Npffr2^{-/-}$ mice, suggesting a gender-specific role for NPFFR2 signalling in the regulation of physical activity during a positive-energy balance that may at least partially contribute to altering energy expenditure in female mice.

We demonstrated that BAT is the critical tissue that acts to reduce energy excess, evidenced by the increased $\Delta T_{BAT-Back}$ under HFD conditions compared with chow. Importantly, our study demonstrates that the full development of a BAT thermogenic response under caloric excess requires NPFFR2 signalling, as a lack of it results in an impaired BAT response accompanied with reduced UCP-1 and PGC-1α expression. In addition to diet-induced thermogenesis, BAT is critical for heat production during cold exposure[49]. Interestingly, our results show that NPFFR2 signalling is not involved in this aspect of BAT function, as mice that lack NPFFR2 signalling maintain a normal BAT thermogenic response to cold exposure. In contrast, NPFFR2 signalling maintains its influence on diet-induced thermogenesis while under altered thermal conditions. This suggests that diet or caloric excess and coldness may induce BAT thermogenesis via different pathways, with NPFFR2 signalling influencing the former but not the later.

Energy homoeostasis regulatory pathways to BAT have been identified to relay from the Arc via the PVN directly projecting to the intermediolateral cell column of the spinal cord[50,51], with another one relaying from the PVN to the locus coeruleus and brain stem to control BAT function[38]. In particular, the later one is a key factor where Arc NPY neurons exert an inhibitory tone on PVN TH neurons subsequently reducing sympathetic outflow to BAT[38]. Considering that NPFFR2 expression is minimal in the PVN but abundant at the Arc[25,32], Arc NPY is likely the key loci in the Arc NPY-PVN-TH pathway for NPFFR2's action. Importantly, NPFFR2 signalling has a crucial influence on Arc NPY expression as a lack of NPFFR2 signalling results in a strong downregulation of Arc *Npy* mRNA rendering this regulatory pathway unresponsive to HFD. As such, this suggests that NPFFR2 signalling is necessary for maintaining levels of NPY required for a balanced energy homoeostatic state and that it also allows for the initiation of an adaptive response when energy balance is challenged such as by caloric excess.

Our results and those from others[41,42] demonstrate the colocalization of NPFFR2 receptors on NPY neurons, suggesting a direct action of NPFFR2 signalling on NPY levels. The effects of NPFF and related peptides on neurons have been demonstrated in electrophysiology studies using various neuronal preparations such as isolated spinal cord tissue[52], dorsal raphe nucleus neurons[53,54] and hypothalamic slice preparations[55]. As NPFFR2 is predominantly coupled to $G_i$ proteins as demonstrated by studies using electrophysiological and in vitro assay approaches[56,57], using pCREB as a marker for forskolin-induced

cAMP activation, we demonstrated that NPFF co-administered with forskolin significantly reduced forskolin-activated NPY neuronal number in the Arc, providing the first evidence for a direct action of NPFFR2 signalling on hypothalamic NPY neurons. However, *Npffr2* single-labelled Arc neurons were also identified by our RNAscope experiments and the co-administration of NPFF with forskolin reduced forskolin-activated Arc NPY neuronal number by approximately 20%. These results leave room for an additional indirect mechanism whereby NPFFR2 signalling controls Arc NPY neurons. Interestingly, Kisspeptin, which has been suggested to act on NPFF receptors[33], was shown to exert an indirect action on Arc NPY neurons that is primarily dependent on GABAergic input to NPY cells[20]. In line with an action of NPFF system through GABAergic transmission, NPFF has been shown to have a dis-inhibitory role via an attenuation of GABAergic inhibitory input to the parvocellular neurons of the PVN in the hypothalamic slice preparation[55]. Furthermore, an indirect action of NPFFR2 signalling on Arc NPY neuronal activity via GABAergic input is also in line with single-cell transcriptome data showing abundant *Npffr2* expression in hypothalamic GABA-expressing non-NPY cells[42]. Thus, it is conceivable that NPFFR2 signalling has dual actions, i.e., direct and indirect, on Arc NPY neurons to regulate diet-induced thermogenesis and BAT function.

Analysis of our conditional KO mouse model with NPFFR2 specifically deleted in NPY neurons demonstrates that specific functions of NPFFR2 signalling depend on a direct action of NPFFR2 signalling on NPY neurons. Specifically, lack of NPFFR2 in NPY neurons shows a transient impairment in diet-induced thermogenesis and BAT thermogenic activity, suggesting that the direct action of NPFFR2 signalling on NPY neurons is especially important in the initiation of diet-induced thermogenesis, but may contribute less to the maintenance of these effects during prolonged energy excess. In addition, our conditional NPFFR2 KO model revealed an important role for direct NPFFR2 signalling on NPY neurons in the control of oxidative fuel selection under obesogenic conditions, which may be counteracted by indirect actions of NPFFR2 signalling on NPY neurons, as these effects were not observed in our germline NPFFR2 KO mice. Moreover, the unaltered food intake observed in conditional NPFFR2 KO mice—in contrast to the reduced food intake in germline NPFFR2 KO—suggests that NPFFR2 signalling does not regulate feeding through direct activity on NPY neurons. Finally, results from the conditional NPFFR2 KO mice demonstrate a critical role for direct NPFFR2 signalling on NPY neurons in the control of bone homoeostasis making this an important control centre for determining energy partitioning.

NPFFR2 receptors can be activated by several ligands and we show here that they have distinct distribution patterns and that their mRNA expression levels are differentially regulated by

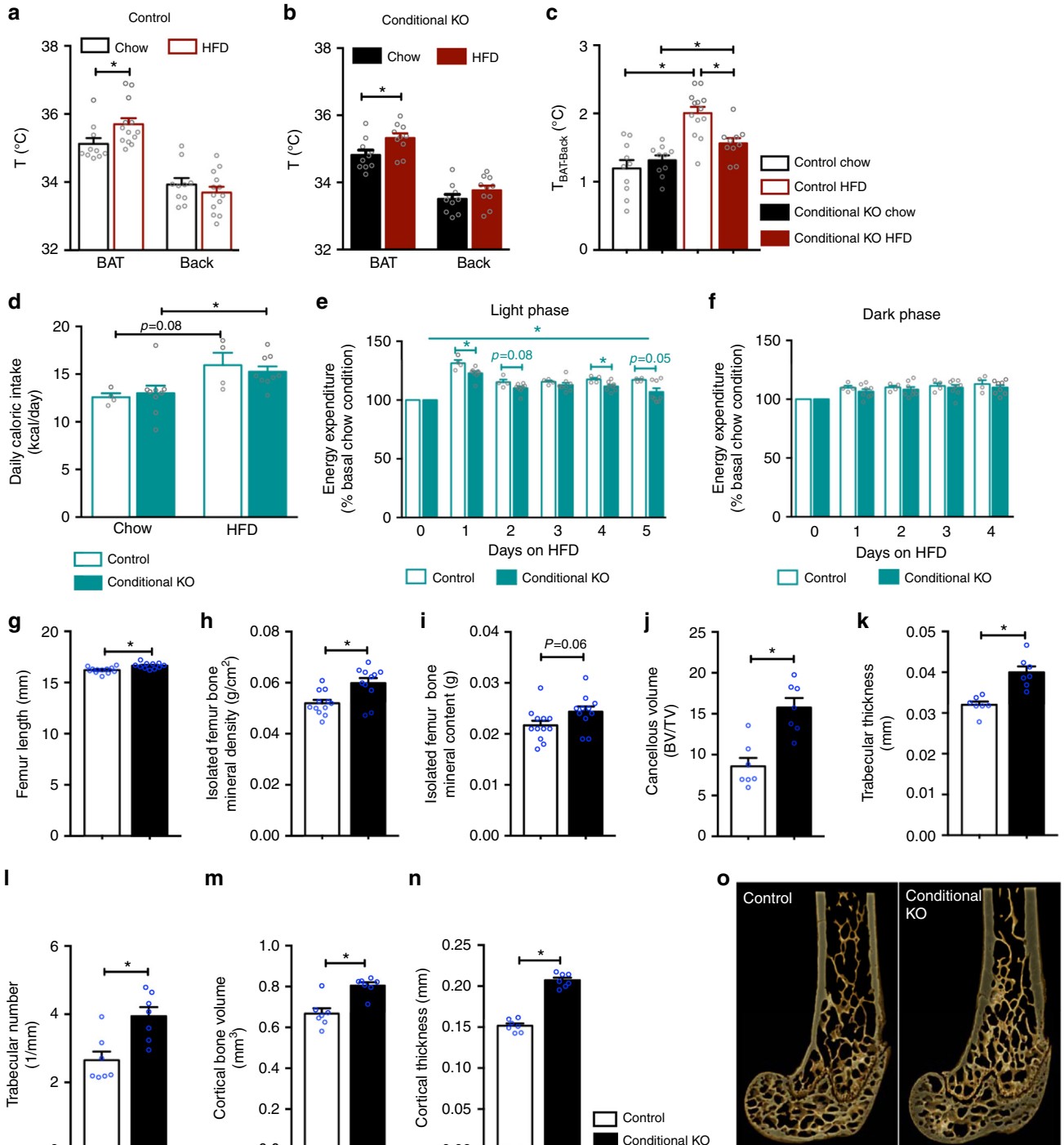

**Fig. 9** Impaired diet-induced thermogenesis in male mice with conditional deletion of *Npffr2* in *Npy* neurons during the early stage of HFD. **a**, **b**, **c** Skin temperature at lumbar back ($T_{Back}$) and brown adipose tissue ($T_{BAT}$), and temperature difference ($\Delta T_{BAT-Back}$) between $T_{BAT}$ and $T_{Back}$ assessed by infrared imaging in freely moving control and conditional KO mice on chow or 10 days of HFD. Control: chow $n = 11$, HFD $n = 13$; Conditional KO: chow $n = 10$, HFD $n = 10$. **d** Energy intake in control and conditional KO mice under chow and during the first 5 days of HFD. Control $n = 4$, conditional KO $n = 8$. **e**, **f** Average energy expenditure during the light and dark phases over the first 5 days of HFD relative to chow condition in control and conditional KO mice. *N* values as in **d**. **g**–**i** The length, bone mineral density and content of isolated femora of control and conditional KO mice at 16 weeks of age. Control $n = 12$, conditional KO $n = 11$. **j**, **k**, **l** Cancellous bone volume, trabecular thickness and trabecular number of the distal femur in control and conditional KO mice. $N = 7$ per group. **m**, **n** Cortical bone volume and cortical thickness of the distal femur in control and conditional KO mice. $N = 7$ per group. **o** Representative photomicrographs of the distal femoral metaphysis in control and conditional KO mice. Gene deletion was induced by tamoxifen at 6 weeks of age (Conditional KO). Male *Npy*$^{creER2/+}$,*Npffr2*$^{lox/lox}$ mice without tamoxifen and *Npy*$^{creER2/+}$,*Npffr2*$^{lox/+}$ mice treated with tamoxifen were used as controls. Data are mean ± SEM. Repeated-measures (**e**, **f**) or one-way ANOVA (**a**–**d**, **g**–**n**) was used to determine difference among groups. **$p < 0.05$ as indicated by bar

energy status. The strong NPFFR2 expression in the Arc shown in our study is in line with anatomical and biological evidence that suggests direct actions of various RFamide peptides towards NPFFR2 in the Arc. Thus, NPVF fibres from the DMH have been shown to have direct projections to Arc[32]. *Kiss1* mRNA and kisspeptin fibres are present in the Arc and can activate Arc neurons via NPFFR2 signalling[20,33]. Moreover, the main NPFFR2 endogenous ligand, NPFF, has been suggested to act on NPFFR2 in a hormone-like fashion due to its existence in the cerebrospinal fluid and plasma, and thus has easy access to the Arc[34,35]. Our study extends these previous reports demonstrating a direct neuronal input from the brain stem neurons to the Arc by showing specific labelling of neurons and fibres in the AP, SubP and NTS upon the delivery of retro beads into the Arc, in support of a possible direct action of NPFF at Arc NPFFR2.

Our results showing that *Npff* expression in the SubP area is upregulated in *Lep*[−/−] mice are consistent with the observed downregulation of NPFF in HFD-fed animals where leptin levels are increased. Although this suggests a role for leptin in the regulation of *Npff* expression in the SubP area, this action is likely to be indirect since little colocalization between *Lepr* and *Npff* was identified. These data also suggest that in addition to the well-known direct action of leptin on Arc NPY neurons, leptin may impact on brain stem NPFF and, in turn, Arc NPFFR2 signalling to regulate Arc NPY neurons that control energy homoeostasis. Possibilities also exist for other factors to control NPFF levels and subsequent NPFFR2 signalling. This is supported by our finding that impaired leptin signalling may not contribute to the exacerbated diet-induced obesity and impaired diet-induced BAT thermogenesis in *Npffr2*[−/−] mice, as HFD did not impair pSTAT3 activation in response to exogenous leptin in these mice.

The location of NPY neurons in the Arc of the hypothalamus, which has been shown to be accessible to circulating factors due to a semi-permeable blood–brain barrier, makes them the ideal coordination centre for receiving signals that communicate energy levels, as seen with factors such as leptin, ghrelin and satiety hormones such as GLP-1 or PYY. However, results from this study showing that NPFFR2 signalling also controls these NPY neurons highlights that additional central neuronal pathways are also important in the control of energy homoeostasis, particularly those that influence energy expenditure and energy partitioning by promoting increased thermogenesis and formation of bone. This seems to be especially critical in the situation of negative energy balance where the levels of peripheral signals are generally low and no longer able to influence NPY levels. The consequence of such a rise in NPY being a strong reduction in both thermogenesis and bone formation, while fat accumulation is promoted. The greater bone mass observed in *Npffr2*[−/−] mice is consistent with the low Arc *Npy* mRNA level, which reflects a positive-energy balance and has been shown to enhance osteoblast activity and bone mass[39]. These results therefore suggest that NPFFR2 signalling controls neuronal NPY populations that influence both the control of thermogenesis as well as bone homoeostasis, two processes that are highly energy consuming. Thus, it may be postulated that NPFFR2 signalling channels excess energy between thermogenesis and bone formation with a lack of the NPFFR2 signalling shifting the balance towards bone anabolism with reduced effects on thermogenesis. As such, NPFFR2 signalling in NPY neurons seems to be critical to switch between these anabolic and catabolic stages by controlling the level of NPY expression and thereby influencing the development of an obese phenotype. As such, targeting this pathway could be a novel and powerful way to control thermogenesis and energy partitioning, thereby providing a potential new treatment option for obesity.

## Methods

**Animals**. All research and animal care procedures were approved by the Garvan Institute/St. Vincent's Hospital Animal Ethics Committee and the University of New South Wales Animal Care and Ethics Committee in accordance with the Australian Code of Practice for the Care and Use of Animals for Scientific Purpose. Mice were housed under conditions of controlled temperature (22 °C for standard laboratory temperature; 28 °C for thermoneutrality) and illumination (12 h light cycle, lights on at 07:00 h). Mice had free access to water and were fed either a normal chow diet (8% calories from fat, 21% calories from protein, 71% calories from carbohydrate, 2.6 kcal/g; Gordon's Speciality Stock Feeds, Yanderra, NSW, Australia) or a HFD (23% calories from fat, 19.4% calories from protein, 4.7% calories from crude fibre, 4.7% calories from acid detergent fibre and 20 MJ/kg; Specialty Feeds, Glen Forrest, WA, Australia).

The generation of *Npffr2*[−/−] mice is detailed in Supplementary Fig. 1. Briefly, a 129/SvJ mouse genomic BAC library (Genome Systems, St Louis, USA) was screened using a mouse *Npffr2* cDNA probe. Positively hybridizing clones were isolated and mapped. Three BamHI fragments covering Exon 2 and 3, as well as surrounding areas, were subcloned and used to generate the targeting construct. A cassette containing the neomycin resistance gene (*Neo*) flanked on one side by a 34 bp-long Cre-recombinase recognition (loxP) site was placed downstream of exon 2 of the *Npffr2* gene. A second loxP sequence was inserted upstream of exon 2 by cloning two complementary 46 mer oligonucleotides into unique NheI site 2.5 kb upstream of Exon 2. Exon 2 contains the translational start of the *Npffr2* gene, which when deleted by Cre-recombinase action results in a non-functional *Npffr2* gene. The linearized targeting vector was transfected into ES cell line Bruce4 (C57BL/6-Thy1.1 background, Ozgene) and positive clones screened by Southern blotting. Two positive clones were injected into C57BL/6 blastocysts and chimeric mice were bred with C57BL/6 mice to generate heterozygous and subsequently homozygous NPFFR2 conditional mice. Crossing of the homozygous NPFFR2 conditional mice with an oocyte-specific Cre-transgenic line was used to generate the germline deleted NPFFR2 KO mice.

To generate an inducible and conditional NPFFR2 KO mouse model where NPFFR2 is specifically deleted from NPY neurons upon tamoxifen induction, *Npffr2*[lox/lox] mice were crossed with *Npy*[CreERT2/+] mice[38] on a C57BL/6 background to generate *Npy*[CreERT2/+];*Npffr2*[lox/+] mice and a breeding colony was maintained by mating *Npy*[CreERT2/+];*Npffr2*[lox/+] and *Npffr2*[lox/lox] mice. Genotyping was performed by PCR using genomic DNA isolated from tail tips. Induction of NPFFR2 deletion from NPY neurons was induced by feeding the *Npy*[CreERT2/+]; *Npffr2*[lox/+] mice with tamoxifen diet (500 mg tamoxifen/kg chow, Specialty Feeds, Glen Forrest, WA, Australia) (referred as 'conditional KO' herein) at 6 weeks of age for 3 weeks after which mice were put back on normal diet for 2 weeks before phenotyping. *Npy*[CreERT2/+];*Npffr2*[lox/lox] mice not receiving tamoxifen diet and *Npy*[CreERT2/+];*Npffr2*[lox/+] receiving tamoxifen were used as controls. Male mice were used for this study due to possible profound impacts of tamoxifen—a selective oestrogen-receptor modulator—on female mice.

For the ribosome translation affinity purification approach (TRAP), ROSA26CAGGFP-L10a mice were sourced from Jax laboratory (B6;129S4-*Gt (ROSA)26Sor*[tm9(EGFP/Rpl10a)Amc]/J) and crossed onto our NPYcre line obtained from Jax laboratory (Tg(Npy-cre)RH26Gsat)[40,58].

**Mice monitoring and tissue analyses**. Body weight of mice was monitored weekly. WT and *Npffr2*[−/−] mice of both genders at 12 weeks of age were examined for whole-body fat and lean masses using DEXA (Lunar PIXImus2 mouse densitometer; GE Healthcare, Waukesha, WI, USA) under light isoflurane anaesthesia. The head and the tail were excluded from DEXA analyses. Spontaneous daily food intake was examined at 15 weeks of age where mice were transferred from group housing on soft bedding to individual cages with paper towel bedding and allowed to acclimatize for two nights. Basal daily food intake was determined as the average of duplicate readings taken over 2 consecutive days. At 18 weeks of age, animals were investigated for energy metabolism using indirect calorimetry as described below. Mice were culled when 20 weeks old between 1200 and 1500 h by cervical dislocation followed by decapitation. White adipose tissue depots (inguinal, epididymal, mesenteric and retroperitoneal) were removed and weighed as published previously[59]. Brain and BAT were removed and frozen for further analysis by quantitative real-time PCR and western blotting, respectively, as described below. Femora were excised, fixed overnight in 4% paraformaldehyde (PFA) in phosphate-buffered saline (PBS) at 4 °C and then stored in 70% ethanol at 4 °C before undergoing processing as detailed below.

Conditional KO mice and control mice were monitored for weekly body weight and examined for spontaneous food intake, energy metabolism by indirect calorimetry with subsequent body composition by DEXA and adiposity by tissue dissection at 14, 15 and 16 weeks of age, respectively, following procedures described above or detailed below. Femora was collected at the tissue collection and processed as described above.

**High-fat feeding study**. A separate set of *Npffr2*[−/−] and WT mice at 8 weeks of age were fed with a HFD (23% calories from fat, 19.4% calories from protein, 4.7% calories from crude fibre, 4.7% calories from acid detergent fibre and 20 MJ/Kg; Specialty Feeds, Glen Forrest, WA, Australia) for 8 weeks. Body weight was monitored throughout the HFD feeding period. After 6 weeks on HFD, mice were

examined for body composition by DEXA scan as described above. After 7 weeks on HFD, metabolic rate, RER ratio and physical activity were examined as described below with concomitant measurement of food intake. Tissue analysis was performed at the end of 8 weeks of HFD feeding as described above.

A subset of conditional KO and control mice at 12.5 weeks of age were fed with a HFD for 5 weeks, with weekly body weight being monitored. Spontaneous food intake, energy metabolism by indirect calorimetry with subsequent body composition analysis by DEXA and adiposity by tissue dissection was performed at 3, 4 and 5 weeks after the commencement of HFD feeding, respectively.

**Indirect calorimetry and measurement of physical activity**. Metabolic rate was measured by indirect calorimetry using an eight chamber open-circuit calorimeter (Oxymax Series; Columbus Instruments, Columbus, OH, USA) as described previously[59]. Briefly, mice were housed individually in specially built Plexiglas cages ($20.1 \times 10.1 \times 12.7$ cm). Temperature was maintained at 22 °C with airflow of 0.6 l/min. Mice were singly housed for at least 3 days before their transfer into plexiglass cages and were acclimatized to the cages for 24 h before recordings commenced. Mice were subsequently monitored in the system for 24 h. Oxygen consumption ($VO2$) and carbon dioxide production ($VCO2$) were measured every 27 min. The RER ratio was calculated as the quotient of $VCO2/VO2$, with 100% carbohydrate oxidation giving a value of 1 and 100% fat oxidation giving value of 0.7[60,61]. Energy expenditure (kcal heat produced) was calculated as calorific value (CV) × $VO2$, where CV is $3.815 + 1.232 \times$ RER[62]. Energy expenditure was normalized to MAT mass (lean body mass + (0.2) × fat mass)[26]. Data for the 24 h monitoring period was averaged for 1 h intervals for energy expenditure and RER. Ambulatory activity of individually housed mice was evaluated within the metabolic chambers using an OPTO-M3 sensor system (Columbus Instruments, Columbus, OH, USA), whereby ambulatory counts were a record of consecutive adjacent photobeam breaks. Cumulative ambulatory counts of X and Y directions were recorded every minute and summed for 1 h intervals.

**Determination of body and BAT temperature**. A subset of male and female WT and $Npffr2^{-/-}$ mice were studied for body and BAT temperatures by a non-invasive high-sensitivity infrared imaging[63,64]. Mice were group-housed throughout the study with two to three mice per cage, unless indicated otherwise, and fed on either chow or a HFD. Two days before infrared imaging, mice were shaved at interscapular and lumbar back regions under light isoflurane anaesthesia to expose the skin of these areas for temperature reading. Infrared imaging experiments were carried out on the following 2 consecutive days and readings from the two measurements were averaged. Briefly, a high-sensitivity infrared camera (ThermoCAM T640, FLIR, Danderyd, Sweden, sensitivity = 0.04 °C) fixed on a tripod was placed 90 cm above the freely moving mice to record the surface temperatures of the mice in a 30–60 s video. Thermo frames that had mouse body naturally extended with both shaved skin regions vertical to camera lenses were extracted from the video. The hottest pixels of the lumbar back and interscapular regions from each extracted frames were obtained simultaneously and used as indicators of body and BAT temperature, respectively.

**Leptin injection and pSTAT3 immunohistochemistry studies**. WT and $Npffr2^{-/-}$ mice at 15 weeks of age were examined for pSTAT3 activation in response to leptin. Leptin (Peprotech #450-31) was administered either via i.p. (2 µg/g body weight) or i.c.v. (1.25 µg, 1 µL) injections with saline-injected mice as control. The i.c.v. injection followed the procedures described previously[65]. Thirty minutes later, under ketamine/xylazine anaesthesia, mice were subjected to cardiac perfusion with saline and 4% formaldehyde in PBS. Brains were removed, post-fixed at 4 °C overnight in 4% formaldehyde in PBS and dehydrated in 30% glucose at 4 °C till sink. Subsequently, brains were stored at − 80 °C until they were sectioned coronally on a cryostat at 40 µm thickness. Free-floating sections were pretreated with citric buffer (pH 6.0) at 80 °C for 30 min, 3% $H_2O_2$ in 50% ethanol for 15 min and blocked in 5% normal goat serum in PBS/0.2% Triton X for 60 min at RT. Anti-pSTAT3 antibody (Cell Signalling #9145, 1:500) was added in antibody diluent containing 2.5% normal goat serum and 0.1% bovine serum albumin (BSA) in PBS/0.2% Triton X and incubated overnight at RT with gentle rocking. Sections were washed, incubated with goat anti-rabbit IgG-biotin secondary antibody (1:250, Sigma #B8895) in antibody diluent for 1 h. Sections were subsequently incubated with ExtrAVidin-Peroxidase (1:250, Sigma #E2886) in antibody diluent for 1 h and the signal was developed by liquid DAB substrate (Dako #K3468). Sections were mounted on Superfrost® slides (Menzel-Glaser, Braunschweig, Germany), coverslipped and photographed with a DM 6000 Power Mosaic microscope (Leica, Germany). To quantify cell numbers, in each mouse ($n = 3–4$), three to four sections between Bregma level − 1.58 mm and − 1.94 mm were chosen. All pSTAT3 immuno-positive cells with a clear profile at Arc were counted and then averaged for each mouse as the number per section.

A separate cohort of WT and $Npffr2^{-/-}$ mice were fed a HFD for 6 weeks and subsequently received i.p. injection of leptin (2 µg/g body weight) or saline (10 µL/g body weight). After 30 min, mice were killed for the immunohistochemical examination of pSTAT3 expression following the protocol described above.

**Western blotting**. Western blotting was performed on BAT samples following procedures described previously[59], in order to determine the protein levels of UCP-1 and PGC-1α. Briefly, BAT samples were resuspended in radio-immunoprecipitation assay buffer (PBS, pH 7.5; 1% nonident NP-40; 0.5% sodium deoxy-cholate; and 0.1% SDS), supplemented with protease and phosphatase inhibitors (10 µg/ml phenylmethylsulfonyl fluoride, 10 µg/ml aprotinin, 10 µg/ml leupeptin, 1 mmol/l $Na_3VO_4$ and 10 mmol/l NaF) and solubilized for 2 h at 4 °C. Equal amounts of tissue lysate (20 µg protein) were resolved by SDS-polyacrylamide gel electrophoresis and immunoblotted with appropriate antibodies against PGC-1α (Millipore AB3242, 1:1000), UCP-1 (Alpha Diagnostics International, UCP11-A, 1:1000) and GAPDH (Cell Signalling #3683, 1:1000). Immuno-labeled bands were imaged using Fusion Fx7 imaging system (Vilber Lourmat, France) and quantitated by densitometry using Fiji: an open-source platform for biological-image analysis. Image of the molecular size marker was overlayed with the image of immunolabeled bands of the same blot to identify specific bands on the blot. The uncropped gel image of the representative bands and the same uncropped gel image overlayed with molecular size marker are shown in Supplementary Figure 9. Protein levels of PGC-1α or UCP-1 were normalized to GAPDH and analysed for per mg protein and per BAT depot as described previously[66], and presented as percentage of that of control group.

**RNA extraction and quantitative real-time PCR**. We measured the mRNA expression levels of $Npvf$ and $Prlh$ in the brain hypothalamic region, and $Npff$ in the brain stem region following procedures described previously[59]. Thus, total RNA from the brain hypothalamic region and brain stem region was extracted using TRIzol® reagent (Sigma, St Louis, MO, USA) following the manufacturer's protocol. The quality and concentration of RNA was determined by measuring the absorbance at 260 and 280 nm using a spectrophotometer (NanoDrop 1000, NanoDrop Technologies, LLC, USA). Two micrograms of total RNA was taken for DNase treatment (RQ1 RNase-Free DNase, Promega, Madison, WI USA) to remove genomic DNA contamination. Subsequently, cDNA was synthesized with oligo(dT)$_{20}$ and random hexamers using Superscript III First-strand Synthesis System for reverse transcription-PCR (Invitrogen, Mount Waverley, VIC, Australia). Quantitative real-time PCR was then carried out on an ABI Prism 7900 HT instrument (Applied Biosystems Inc, Foster City, CA USA) using the TaqMan Universal PCR master mix (Applied Biosystems Inc., Foster City, CA USA) following the manufacturer's instructions. Probes and primers for the target gene, the $Npvf$ (Mm00452052_m1), $Npff$ (Mm00450676_g1), $Prlh$ (Mm01286067_m1) and the reference gene, $Actb$ (Mm00607939_s1), were selected from TaqMan gene expression assay reagents (Applied Biosystems, Inc., Foster City, CA USA). Fluorescent signals generated during PCR amplifications were monitored and analysed with ABI Prism 7900 HT SDS software (Applied Biosystems, Inc., Foster City, CA USA). In order to determine the efficiency of each TaqMan gene expression assay, standard curves were generated by serial dilution of cDNA and quantitative evaluations of target and reference gene levels were obtained by measuring threshold cycle numbers (Ct). The relative expression of target mRNA was computed from the target gene Ct values and the reference gene $Actb$ Ct value using the standard curve method (Sequence Detection Systems Chemistry Guide, Applied Biosystems, Inc). For measuring the expression level of $Npffr2$ and $Kiss1r$, LightCycle (LightCycler 480 Real-Time PCR system; Roche Applied Science, Mannhelm, Germany) was used. The primers 5′- AAGCAGCATGTGCAA-GATCA-3′ and 5′- TGACAGTGAGCTTTGGCTTA-3′ were used for $Npffr2$, 5′-TGCTGGCTCTATATCTGCTG-3′ and 5′-CTTGAAGCACCAGGAACAGC-3′ were used for $Kiss1r$. The PCR condition used in measuring these genes were as following[67]: 94 °C for 30 s, 62 °C for 30 s, 72 °C for 20 s for 40 cycles. Expression of the gene was normalized to the expression of housekeeping gene $Actb$.

**In situ hybridization**. Coronal brain sections (25 µm) were cut on a cryostat and thaw-mounted on Superfrost® slides (Menzel-Glaser, Braunschweig, Germany). Matching sections from the same coronal brain level were assayed together using DNA oligonucleotides complementary to mouse $Npy$ (5′-GAGGGTCAGTCCAC ACAGCCCCATTCGCTTGTTACCTAGCAT-3′), mouse pro-opiomelanocortin ($Pomc$) (5′-TGGCTGCTCTCCAGGCACCAGCTCCACACATCTATGGAGG-3′), mouse thyrotroppin-releasing hormone ($Trh$) (5′-AACCTTACTCCTCCAGAG GTTCCCT GACCCAGGCTTCCAGTTGTG-3′), mouse TH ($Th$) (5′-CTCTAAG GAGCGCCGGATGGTGTGAGGACTGTCCAGTACATCA-3′, mouse $Npff$ (5′-GAGACTGAGGAAGGCACAGGCAAGCAAGGAGCCATGAACCACAGG-3′) and mouse $Prlh$ (5′-GCTGTGAGAGAACTTGGCACTTCCATCCAGTGGGAA GCAGCTTAG-3′). Oligonucleotides were labelled with [α-$^{35}$S] thio-dATP (PerkinElmer, Boston, MA, USA) using terminal deoxynucleotidyltransferase (Roche, Mannheim, Germany) following manufacturer's instructions and purified through a G25 column. Fresh brain sections were fixed in 4% PFA, rinsed twice in PBS, pretreated with acetic anhydride (0.25% vol/vol), rinsed twice in PBS, delipidated in chloroform, dehydrated in ethanols, air dried and hybridized with radiolabelled probe ($2.5–5.0 \times 10^5$ c.p.m. per section) overnight in a 42 °C humidified chamber. The slides were then washed in three times in 5xSSC with 0.5 mM DTT (1,4-Dithiothreitol, VWR International #APLA29480005), twice in $2 \times$ SSC/formamide (50% vol/vol) with 50 mM DTT at 43 °C, cooled to RT, rinsed in $1 \times$ SSC, $dH_2O$ and 70% ethanol, and air dried. Subsequently, hybridization signals on sections were visualized by exposure to BioMax MR film (Kodak, Rochester, NY, USA) and

digitalized images from the scanned autoradiograms were acquired. For structural details, the brain sections were photoemulsion-dipped and superficially counter-stained with haemotoxylin, with which regions of interest were visualized and captured into digital images acquired by bright-field microscope (DM 6000 Power Mosaic microscope, Leica, Germany). Quantification of the mRNA expression levels of respective genes were performed by densitometry for brain areas of interest outlined with consistent defined dimensions across corresponding sections on the photomicrographs using Fiji software for Mac OS.

We used RNAscope duplex chromogenic assay (Advanced Cell Diagnostics, Inc.)—a novel ISH technique to visualize multiple cellular mRNA targets in fresh frozen tissues[68]—to detect colocalisation of *Npffr2* and *Npy* in the Arc, *Npffr2* and *Pomc* in the Arc, *Lepr* and *Npff* in the brain stem. Coronal brain sections (25 μm) were cut on a cryostat and thaw-mounted on Superfrost® slides (Menzel-Glaser, Braunschweig, Germany), and dual labelled for the mRNA of *Npy* (ACD #313321-C2) and *Npffr2* (ACD #410171), *Pomc* (ACD #314081-C2) and *Npffr2*, or *Lepr* (ACD #402731) and *Npff* (ACD #300031-C2), using RNAscope® 2.5 Duplex Detection Kit following manufacturer's protocol (Advanced Cell Diagnostics, Inc.). Section staining was photographed using a DM 6000 Power Mosaic microscope (Leica, Germany).

**pCREB immunohistochemistry and colocalization with NPY neurons**. To investigate the signalling response of NPY neurons to NPFFR2 activation, we examined the effect of NPFFR2 signalling activation by NPFF on forskolin-induced phosphorylation of the CREB—a downstream response to forskoline-induced increases in cAMP level—in NPY-GFP mice[46,47]. NPY-GFP mice at 15 weeks of age received i.c.v. injection of one of the following four solutions at 1 μL: Saline, forskolin (Sigma #F6886, 2.5 mM), NPFF (150 nM) or forskolin (2.5 mM) + NPFF (150 nM), under ketamine/xylazine anaesthesia. NPFF was kindly provided by Professor Annette G Beck-Sickinger (University Leipzig, Germany). Procedures for i.c.v. injection were as described previously[65]. Thirty minutes after the injection, mice were killed by cardiac perfusion with saline and 4% formaldehyde in PBS. Brains were removed, post-fixed at 4 °C overnight in 4% formaldehyde in PBS and dehydrated in 30% glucose at 4 °C till sink. Subsequently, brains were stored at −80 °C till sectioned coronally on a cryostat at 40 μm thickness. Free-floating sections were blocked in 5% normal goat serum in PBS/0.2% Triton X for 60 min at RT and incubated with anti-pCREB antibody (Cell Signalling #9198, 1:800) in antibody diluent containing 2.5% normal goat serum and 0.1% BSA in PBS/0.2% Triton X overnight at RT with gentle rocking. Sections were washed, incubated with donkey-anti-rabbit Cy3 antibody (1:500, Jackson ImmunoResearch Laboratories, Inc. #711-165-152) in antibody diluent for 2 h. Sections then were washed, mounted on Superfrost® slides (Menzel-Glaser, Braunschweig, Germany), cover-slipped and photographed using fluorescent microscope (Leica DM 5500, Germany) equipped with a colour camera (DFC310 Fx). For quantification, two to three sections between Bregma level − 1.70 mm and − 1.94 mm from each mouse (*n* = 5 and 4 for forskolin and forskolin + NPFF group, respectively) were chosen. GFP-labelled NPY neurons, Cy3-labelled pCREB-ir-positive neurons and NPY/pCREB colocalizing neurons in the arcuate nucleus were counted using Fiji software for macOS. The two sides of the arcuate nucleus were counted separately and counts per side of the arcuate nucleus from the same brain were averaged and presented.

**Retrograde neuronal tracing**. WT mice were stereotaxically injected into one side of Arc with 0.5 μl of red retrograde beads (Lumaflour, Inc.). Fourteen days after injection, mice were killed by cardiac perfusion with saline and 4% formaldehyde in PBS. Brains were removed, post-fixed at 4 °C overnight in 4% formaldehyde in PBS and dehydrated in 30% glucose at 4 °C till sink. Subsequently, brains were stored at − 80 °C till sectioned coronally on a cryostat at 40 μm thickness. Free-floating sections were collected and mounted on Superfrost® slides (Menzel-Glaser, Braunschweig, Germany), coverslipped and examined using a Leica DM 5500 fluorescent microscope. Tracer labelling was viewed under fluorescent filter set with 540–552 nm, 565 nm and 580–620 nm for excitation, beamsplitter and emission, respectively. Tracer signal was also examined for possible crossover to other channel. For this we used a green channel filter set with 460–500 nm, 505 nm and 512–542 nm for excitation, beamsplitter and emission, respectively. Sections were photographed using a colour camera (DFC310 Fx) equipped to the microscope.

**TRAP reverse-transcriptase qPCR**. TRAP experiment was performed based on the previously published protocol with some modifications[40,69]. Thus, the Arc was dissected out by careful manual dissection under the inverted microscope using the forth ventricle as the reference in chilled dissection buffer (1 × Hank's balanced salt solution, 2.5 mM HEPES-KOH [pH 7.4], 35 mM Glucose and 4 mM NaHCO₃) supplemented with fresh 100 μg/mL of cycloheximide (CHX; Sigma). The tissue was homogenized with a hand pestle mixer (Argos Technologies) in 500 μL of lysis buffer (20 mM HEPES-KOH [pH 7.4], 5 mM MgCl₂, 150 mM KCl, 0.5 mM DTT) supplemented with fresh protease inhibitor (1 tablet/mL; Roche Mini Complete, EDTA-Free), 100 μg/μL CHX and 40 U/mL of RNAsin (Promega), and then incubated at 4 °C for 5 min. Homogenates were centrifuged for 10 min at 2000 × *g* at 4 °C to remove pellet nuclei and cell debris, and then 50 μL of NP-40 working solution (10% vol/vol; Biochemica) and DHPC (300 mM; Avanti Polar Lipids) were

added to the supernatant and mixed gently by inverting the mixture 10 times. After incubation on ice for 5 min, the lysate was centrifuged for 10 min at 13,000 × *g*. 20% of the lysate was kept as input. Regarding the antibody-beads preparation step, 50 μL protein G Dynal magnetic beads (Invitrogen) were washed three times with 0.15 M KCl buffer (20 mM HEPES KOH [pH 7.4], 5 mM MgCl₂, 150 mM KCl, 1% NP-40, 0.5 mM DTT, 100 μg/ml CHX) at RT and then 10 μL of anti-GFP antibodies (2 μg/μL; Invitrogen A11122) were added into the beads and incubated with beads suspended in 0.15 M KCl buffer (275 μL) for 1 h at RT with mild end-to-end rotation. Next, the antibody-bound beads were collected by using the magnetic rack and washed three times with 0.15 M KCl buffer before use. The beads were then mixed with the cell-lysate supernatant and the mixture was incubated at 4 °C with mild end-to-end rotation overnight. Beads were subsequently collected on a magnetic rack, washed three times with 0.35 M KCl buffer (20 mM HEPES KOH [pH 7.4], 5 mM MgCl₂, 350 mM KCl, 1% NP-40, 0.5 mM DTT, 100 μg/ml CHX) at 4 °C and immediately placed in 350 μL of RNA lysis buffer (RLT) supplemented with 10% of 2-Mercaptoethanol at RT and incubated for 5 min. Next, the RLT-containing RNA was purified with the RNeasy microKit (Qiagen) following the manufacturer's protocol. DNase digestion step was included in the purification. RNA quantitation and purity were confirmed by NanoDrop Spectrophotometers. Forty nanograms of RNA was used to synthesis cDNA by using the SuperScript III First-Strand Synthesis System (Thermo Fisher Scientific). RT-qPCR using primers for *Npy*, GFP and *Npffr2* were carried out in samples prior (input) and after the immunoprecipitation in at least triplicates from 1:5 dilution cDNA from each sample using the LightCycler® (LightCycler® 480 Real-Time PCR system, Roche Applied Science, Germany), SYBR Green I (Molecular Probes) and Platinum Taq DNA Polymerase (Invitrogen). Primers 5′-AATCAGTGTCTCAGGGCTG-3′ and 5′-CTATCTCTGCTCGTGTGTTT-3′ were used for assaying *Npy*, 5′-CGGATC TTGAAGTTCACCTT-3′ and 5′-GAGCGCACCATCTTCTTCA-3′ were used for GFP, 5′-GTACGACCAGAGGCATACA-3′ and 5′-AGCACCCTGTGCTGCTCA-3′ were used for *Actb*. The primers used for *Npffr2* have been described in RNA extraction and quantitative real-time PCR. The PCR condition used in all the TRAP RT-qPCR experiment were as following: 94 °C for 30 s, 62 °C for 30 s, 72 °C for 20 s for 40 cycles[67]. Expression of the gene was normalized to the expression of housekeeping gene *Actb*.

**Bone μCT**. Following fixation, left femora were cleaned of muscle and analysed using μCT with a Skyscan 1172 scanner and associated analysis software (Skyscan, Aartselaar, Blegium), as previously described[70]. Briefly, analyses of the cortical bone were carried out in 229 slices (1.0 mm) selected 572 slices (2.5 mm) proximally from the distal growth plate resulting in calculations of the following parameters: total tissue area, bone area, marrow area, periosteal perimeter, cortical thickness and polar moment of inertia (an index of strength). Analyses of the trabecular bone were carried out in 229 slices (1.0 mm) and selected 114 slices (0.5 mm) proximally from the distal growth plate resulting in calculations of the following parameters: total tissue volume, bone volume, trabecular number, trabecular thickness and trabecular separation.

**Behavioural testing**. WT and *Npffr2*−/− mice at 17 weeks of age were tested in a battery of behavioural tests with an inter-test interval of at least 48 h: elevated plus maze (EPM), open field (OF), prepulse inhibition, motor function tests (i.e., pole test and beam walking test) and hot water tail-flick test[71]. All tests were conducted during the first 5 h of the light phase to minimize effects of the circadian rhythm on the performance of test mice.

*Elevated plus maze*: The EPM assesses the natural conflict between the tendency of mice to explore a novel environment and avoidance of a brightly lit, elevated and open area. The grey plus maze was '+' shaped as described previously[72]. Mice were placed at the centre of the '+' (faced towards an enclosed arm) and were allowed to explore the maze for 5 min. The time spent in and number of entries into the open and enclosed arms were recorded using AnyMaze™ (Stoelting, Wood Dale, USA) tracking software (percentages calculated based on total arm entries/total are time excluding centre time).

*Open field*: The OF test provides measures of locomotion, exploration and anxiety-like behaviour. This test mimics the natural conflict in mice between the tendency to explore a novel environment and to avoid an exposed open area. Test mice were placed for 30 min into an infrared photobeam-controlled OF activity test chamber (43.2 × 43.2 cm—illumination level: 20 lx; MED Associates, Inc., Vermont, USA), which was divided into a central and a peripheral zone (MED software coordinates for central zone: 3/3, 3/13, 13/3, 13/13; box size: 3; ambulatory trigger: 2; resting delay: 1000 ms; resolution: 100 ms). Measures of locomotion included distance travelled and ambulatory time; vertical activity was taken as a measure of exploration (i.e., rearing).

*Hot water tail-flick test*: Pain sensitivity (i.e., nociception) was assessed using the hot water tail-flick test. Mice were gently wrapped in a soft tissue towel and the tail tip was placed in a 52 °C warm water bath and the latency to flick the tail was recorded. Each test consisted of three trials (intertrial interval of 3–4 min). In between trials, mice were returned to their home cage.

**Statistical analyses**. All data are expressed as means ± SEM. Energy expenditure, RER and physical activity over the continuous 24 h period were averaged for the

whole 24 h period, as well as for the light and dark periods. Differences between different genotypes were assessed by two-sided analysis of variance (ANOVA) or repeated-measures ANOVA combined with Tukey's honest significant difference post-hoc analysis where appropriate. Statistical analyses were performed with SPSS for Mac OS X, version 16.0.1 (SPSS, Inc., Chicago, IL, USA). Statistical significance was defined as $p < 0.05$.

**Oligonucleotides**. A list of the oligonucleotides used in this study is presented in Supplementary Table 3.

## Data availability

The data supporting the findings of this study are available within the article and its Supplementary Information files, and from the corresponding author upon reasonable request.

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

## Acknowledgements

We thank the staff the staff of the Garvan Institute Biological Testing Facility, staff of the Australian BioResources and Adam Bryan at Neuroscience Research Australia for facilitation of these experiments and taking care of our test mice. We thank Jerry Tanda for critical comments on the manuscript. We thank Irene Kahr for technical assistance in generating the targeting construct for the *Npffr2* gene. We thank Ananda Kalutantiri for the information technology support. This research was supported by the National Health and Medical Research Council of Australia (NHMRC) with project grants (#1102012), a Career Development Fellowship (#1045643) to T.K. and a Research Fellowship to H.H. (#1118775). T.K. was also supported by the NHMRC dementia research team initiative (#1095215) and the Rebecca L. Cooper Medical Research Foundation.

## Author contributions

L.Z. and H.H. designed and supervised the project. L.Z., C.K.I., I.J.L., F.R., Y.Q., T.K., J.K.L., N.J.L. and R.F. performed experiments and acquired the data. L.Z., T.K., P.A.B. and H.H. analysed and interpreted the results. L.Z. and H.H. wrote the manuscript.

## Additional information

**Competing interests:** The authors declare no competing interests.

