## [Peer Review File · Nature Communications]

Reviewers' Comments:

Reviewer #1:

Remarks to the Author:

Using loss-of-function genetic approaches, the authors determined a role for neuropeptide FF receptor 2 (NPFFR2) in the regulation of energy balance in mice. They revealed the gender-dimorphic effects of NPFFR2 on food intake, energy expenditure and bone mass in mice fed regular chow. Thus, a loss of NPFFR2 signaling results in reduced food intake and body weight without altering energy expenditure and physical activity in male mice, whereas in females such deficiency increases energy expenditure and physical activity and lowers fat mass with no feeding effect. A lack of NPFFR2 signaling also leads to increased cancellous bone in males and cortical bone in females. In addition and most importantly, they ascertained a similar effect of NPFFR2 on diet-induced thermogenesis in both genders. While high fat diet results in increases in energy expenditure and BAT temperature in wild-type mice, a loss of NPFFR2 signaling attenuates these increases. A loss of NPFFR2 signaling does not affect cold-induced thermogenesis. Last, they established that NPFFR2 acts in diet-induced adaptive thermogenesis via an Arc NPY signaling pathway based on the observations that Npy mRNA expression is decreased in the Arc of NPFFR2 knockout mice, HFD does not cause a further reduction of Arc Npy expression in NPFFR2 knockout mice, a subset of NPY neurons expresses NPFFR2, NPFF reduces forskoline-induced pCREB in NPY neurons, and NPFFR2 deletion specifically in NPY neurons alters the effects of HFD on food intake, oxygen consumption and bone mass. Together, the authors claimed that diet-induced adaptive thermogenesis requires NPFFR2 signaling. Diet-induced adaptive thermogenesis was reported years ago but how the brain acts in this adaptation remains incompletely understood. The research approach in this report is sound, and the work is timely providing neural evidence underlying diet-induced thermogenesis. However, there are some flaws in the paper that weaken their proclamation of NPFFR2-NPY signaling regulation of diet-induced thermogenesis.

- 1) Figure 1 shows alterations in mRNA expression of the NPFFR2 ligands NPFF, NPVF and PrRP in the brainstem and/or the hypothalamus in mice under different feeding conditions (HFD vs. fast) as well as in ob/ob mice. Since the paper focuses on NPFFR2, it would be better to include the results showing whether brain NPFFR2 expression is also altered in these test conditions.
- 2) Both physical activity and BAT thermogenesis contribute to energy expenditure. NPFFR2 knockout females on chow have increased physical activity and energy expenditure (Figures 3b, d, j), but no alterations in BAT temperature (Figures 4a-b), suggesting that physical activity may play an important role in the effect of NPFFR2 on energy expenditure. Consistent with this view, decreased physical activity in female NPFFR2 knockout mice on HFD is also likely a contributing factor to their reduced energy expenditure (Figures 3f-j). Such effects need to be addressed in the paper in addition to BAT thermogenesis, providing the readers with a full understanding of how NPFFR2 acts in the regulation of energy expenditure.
- 3) On page 11 lines 6-8, they described the results that HFD-induced increases in BAT temperature in WT mice were not seen in NPFFR2 knockout mice (Figure 4d), which actually showed a trend to decreased temperature. To me, this is likely due to a higher level of BAT temperature (more than 36°C) in this cohort of NPFFR2 knockout mice on chow compared to data in Figure 4a. Are there any environmental (such as stress) or body weight differences between these two tests?
- 4) The quality of Western blot image is low (Figure 4i), for instance, the loading control GAPDH was very weakly detected and the samples were unevenly loaded based on the levels of GAPDH. UCP-1 and PGC-1 α levels should be normalized by the loading control GAPDH for group comparisons.
- 5) On page 14 lines 9-11, they stated that "The POMC mRNA expression in the ARC was significantly increased in response to HFD in both WT and NPFFR2 knockout mice without significant differences between genotypes (Fig. 5e)". However, the figure shows that ARC POMC mRNA expression was significantly increased only in NPFFR2 knockout mice on both diets. Please clarify.
- 6) In the study of NPFFR2 deletion specifically in NPY neurons, they only showed the effect of this deletion on oxygen consumption in Figure 7. It would be better to present energy expenditure data

as shown in Figure 3 for comparisons. Also, it would be better to specify whether the results were obtained from male mice, females or both.

7) Last, they report that “lack of NPFFR2 signaling on NPY neurons caused significant increases in femur length, femur BMD and a trend to an increase in femur BMC (Fig. 7h-j) compared to control mice”, but not provide the gender information. For me, this information is important because of the gender-dimorphic effect of global NPFFR2 knockout on bone mass, i.e., decreased femoral length in males (Fig. 6a) and unaltered femoral length in females (Fig. 6e). Particularly, these results from global knockout are not consistent with the finding of increased femoral length in mice with NPFFR2 deletion in NPY neurons. Thus, any similarities or discrepancies between NPY-neuron-specific and global NPFFR2 knockout need to be addressed and discussed.

Reviewer #2:

Remarks to the Author:

General comments:

The authors of this paper explore the roles of the neuropeptide FF receptor 2 (NPFFR2) in the regulation of body weight (adiposity, energy expenditure, activity, food intake) and bone growth. They conclude that lack of NPFFR2 results in body-weight gain in mice fed a high-fat diet due to decreased energy expenditure as a result of decreased activation of brown fat thermogenesis. They provide some evidence that NPFF acting directly on NPY-expressing neurons is responsible for the phenotype of the knockout mice. The paper includes a wealth of data that, in general, are clearly described. Nevertheless, there are a number of problems that should be addressed.

1. If the thesis is that NPFF-expressing neurons in the NTS (sub-AP) project axons to the arcuate nucleus to act on NPFFR2 receptors on NPY-expressing neurons, then including experiments showing this connectivity is important. Showing that NPFFR2 is co-expressed with NPY in the Arc is a start, but is NPFF normally released in the vicinity of NPY neurons in the Arc?
2. It is unclear why the authors focused on NPFFR2 rather than NPFFR1 (or both). Is NPFFR1 also expressed in NPY neurons in the Arc? Is NPFFR1 activated by the same ligands as NPFFR2? Could inactivation of NPFFR2 be compensated for by NPFFR1? Does Kiss1 (made by neighboring neurons) also regulate NPFFR2 like it does NPFFR1?
3. Does NPFF injection into the Arc of WT and NPFFR2 KO (or the conditional KO) mice reduce food intake by hungry mice.
4. How efficient is the conditional knockout? How long does it take for inactivation to occur?
5. It is clear that mice can compensate for the loss of NPY/AgRP neurons, hence it follows that they can compensate for loss of any critical signaling molecule made by those neurons. Thus, the conditional knockout of NPFFR2 in adult mice is an important experiment. However, to conclude that NPY neurons mediate most of the effects of the constitutive knockout, the critical experiments shown in Fig. 2-4 should be replicated with the conditional KO. The time from when tamoxifen was added to the chow until experiments were performed is not stated; it should match that of the measurements in Fig. 2. The gender of the mice studied in Fig. 7 should be mentioned. What is the effect of the conditional knockout on BAT temperature, food intake, adiposity, insulin, leptin, etc.
6. It seems odd that with constitutive knockout the major effect of NPFFR2 inactivation on energy expenditure is at night, whereas with the conditional inactivation it is during the day. What is the explanation for this.
7. NPY is expressed broadly in the brain, so it is not clear that the conditional knockout of NPFFR2 in NPY neurons is having its effect primarily on the arcuate population. Conditional knockout in AgRP neurons would be more meaningful.
8. The experiment in Fig. 7d should be quantified. How much is pCREB reduced in NPY neurons? It looks like pCREB was reduced in non-NPY neurons in the Arc as well; that should also be quantified.

Overall, the observation that NPFFR2 regulates thermogenesis, which affects energy expenditure and body weight is interesting. A paper more focused on the circuitry involved would be appreciated. Some of the other data (e.g. bone growth) detracts from these central questions.

Editorial comments:

1. Approved mouse gene names in italics should be used when referring to genes and their mRNAs. All mouse gene names start with a capital letter followed by lowercase letters and numbers. Ob/ob is not a gene name, it is an allele of Lep gene.
2. Authors should consistently use hyphens when using nouns as adjectives: a few examples, high-fat diet, neuropeptide-expressing neurons, ad libitum-fed mice, activity related alterations, metabolism-related phenotypes, tail-flick test. p 11, No hyphen "after 6 weeks of chow"
3. Always leave a space between numbers and their units, e.g. 24 h (especially bad in Methods)
4. Use em-dash appropriately (not an comma and a hyphen)
5. The genetic background of mice should be indicated.
6. Approval of animal experiments should be included.
7. P 6, "8 weeks of feeding" no hyphen, spelling
8. P 6, changes.....suggest (not suggests)
9. P 13, "these results show..." not showing
10. ISH does not need to be in italics.
11. P 17, NPFFR2 lox/lox, flox/flox , flox/fox; what is the distinction between these different designations?
12. P 19, Is "overwriting" meant or should this be overriding?
13. P 21, colocalization (spelling)

Reviewer #3:

Remarks to the Author:

This study evaluates the role of neuropeptide FF receptor 2 (NPFFR2) in energy balance. The manuscript suggests that NPFFR2 plays a critical role in NPY neurons to control energy expenditure and BAT activity particularly when mice are fed a HFD. Although the hypothesis is interesting, there are important issues on this study:

1. Figure 5. These results demonstrate that NPFFR2 signalling regulates not only Arc NPY levels but also POMC levels. It is surprising that the authors just discard the potential role of NPFFR2 signalling on POMC neurons, which are also relevant in the control of thermogenesis and whole body energy homeostasis. This aspect is of particular relevance since there are also NPY neurons that do not express NPFFR2 (Fig. 7C-ii) as well as non-NPY neurons that do express NPFFR2 (Fig. 7C-iii), and as the authors state, this suggests that NPFFR2- signalling could also affect other Arc neuronal pathways. Therefore, to do a full characterization of the role of NPFFR2- signaling in POMC neurons is important. Moreover, from this figure it is not possible to conclude that "through that influences the responsiveness of Arc NPY neurons to a HFD that subsequently results in impaired diet-induced BAT thermogenesis in NPFFR2-/- mice."
2. Another key aspect would be to measure the electrical activity of NPY (and POMC) neurons.
3. NPFFR2 is expressed in other tissues besides the brain (i.e. adipose tissue macrophages, see PMID: 28581443) and seems to be metabolically important. Thus, studies using NPFFR2-/- mice should be taken with caution here, and complete studies in conditional KO models are essential to provide convincing results on the specificity of hypothalamic NPFFR2
4. The characterization, or at least the description here, of mice lacking NPFFR2 in NPY neurons is very succinct. For example, data on food intake, body composition and temperature, which are from my point of view essential in this study, are not shown. In addition, the phenotype of these mice does not entirely recapitulate the findings observed in the global KO. For instance, in conditional KOs, daily caloric intake is identical between genotypes when fed a HFD while it was different in global KO (Fig 2i), suggesting that the effects of NPFFR2 signaling on feeding may be independent of NPY, and this may take us again to POMC neurons.
5. To fully characterize the specificity of the results, over-expression in NPY and POMC neurons is advisable.

Minor:

Reviewer #4:

Remarks to the Author:

Zhang et al` s manuscript describes a new signalling pathway involving NPFF receptor 2 (NPFFR2) that controls food intake and energy expenditure by promoting diet-induced thermogenesis in the brown adipose tissue (BAT). The authors provide convincing results that NPFFR2 is required to control energy expenditure and that the signaling pathway activating BAT thermogenesis is initiated in NPY neurons. The results are novel and of high interest in the field. The experimental approaches and the statistical analysis are of high standard. At the end of the manuscript they used Translating Ribosome Affinity Purification Approach to provide evidences that NPFFR2 signaling acts directly on NPY neurons. For this they generated mice that express a GFP translational fusion of the ribosomal protein L10 (L10a-GFP), refereed to as NPY-TRAP, by crossing TRAP lox/lox mice with NPY CRE/+ mice. The authors verified accumulation of L10a-GFP in the Arcuate nucleus by using fluorescent microscopy. Upon immunoprecipitation (IP) with antiGFP antibodies, the authors showed that L10a-GFP transcripts were present in the IPs of NPY-TRAP mice , but not in the Ips of Wt mice. These IPs were enriched in NPY mRNAs as compared with IP of WT mice. NPFFR2 was also found enriched in the IPs from NP-Y neurons. At this point, it is my impression that it is important to show the mRNA levels of NP-Y, L10a-GFP, and NPFFR2 in IPs obtained from the parental TRAP lox/lox and NPY Cre/+ mice instead of in WT mice to straighten the conclusion of the experiment. To provide evidence of enrichment they can also compare IP of NPY-TRAP mice with IP from mice that ubiquitously express L10a-GFP. This is of particular importance since the authors later showed, by in situ hybridization, that there is juts partial co-expression of NP-Y and NPFFR2.

Responses to Reviewers' comments

Reviewer #1 (Remarks to the Author):

Using loss-of-function genetic approaches, the authors determined a role for neuropeptide FF receptor 2 (NPFFR2) in the regulation of energy balance in mice. They revealed the gender-dimorphic effects of NPFFR2 on food intake, energy expenditure and bone mass in mice fed regular chow. Thus, a loss of NPFFR2 signaling results in reduced food intake and body weight without altering energy expenditure and physical activity in male mice, whereas in females such deficiency increases energy expenditure and physical activity and lowers fat mass with no feeding effect. A lack of NPFFR2 signaling also leads to increased cancellous bone in males and cortical bone in females. In addition (Liu and Herbison, 2015) and most importantly, they ascertained a similar effect of NPFFR2 on diet-induced thermogenesis in both genders. While high fat diet results in increases in energy expenditure and BAT temperature in wild-type mice, a loss of NPFFR2 signaling attenuates these increases. A loss of NPFFR2 signaling does

not affect cold-induced thermogenesis. Last, they established that NPFFR2 acts in diet-induced adaptive thermogenesis via an Arc NPY signaling pathway based on the observations that Npy mRNA expression is decreased in the Arc of NPFFR2 knockout mice, HFD does not cause a further reduction of Arc Npy expression in NPFFR2 knockout mice, a subset of NPY neurons expresses NPFFR2, NPFFR2 reduces forskoline-induced pCREB in NPY neurons, and NPFFR2 deletion specifically in NPY neurons alters the effects of HFD on food intake, oxygen consumption and bone mass. Together, the authors claimed that diet-induced adaptive thermogenesis requires NPFFR2 signaling. Diet-induced adaptive thermogenesis was reported years ago but how the brain acts in this adaptation remains incompletely understood. The research approach in this report is sound, and the work is timely providing neural evidence underlying diet-induced thermogenesis. However, there are some flaws in the paper that weaken their proclamation of NPFFR2-NPY signaling regulation of diet-induced thermogenesis.

We thank the reviewer for the positive and encouraging comments about our study.

1) *Figure 1 shows alterations in mRNA expression of the NPFFR2 ligands NPFF, NPVF and PrRP in the brainstem and/or the hypothalamus in mice under different feeding conditions (HFD vs. fast) as well as in ob/ob mice. Since the paper focuses on NPFFR2, it would be better to include the results showing whether brain NPFFR2 expression is also altered in these test conditions.*

Our response: Thank you for your suggestion. From our experience with neuropeptide systems it is normally the ligands that are strongly regulated by alterations in energy status with rather small to no effect in the corresponding receptors. Nevertheless, we have performed quantitative RT-PCR to assess the hypothalamic expression of *Npffr2* under both normal chow-fed conditions and HFD conditions in WT mice. Relative to chow-fed conditions, HFD feeding resulted in a down-regulation of hypothalamic *Npffr2* expression, which is consistent with the reduced *Npff* expression under this condition and suggests an overall reduced NPFFR2-ergic tone in response to a positive energy balance. We have presented these new results in **Fig. 5d** and discussed the results in our revised manuscript.

2) *Both physical activity and BAT thermogenesis contribute to energy expenditure. NPFFR2 knockout females on chow have increased physical activity and energy expenditure (Figures 3b, d, j), but no alterations in BAT temperature (Figures 4a-b), suggesting that physical activity may play an important role in the effect of NPFFR2 on energy expenditure. Consistent with this view, decreased physical activity in female NPFFR2 knockout mice on HFD is also likely a contributing factor to their reduced energy expenditure (Figures 3f-j). Such effects need to be addressed in the paper in addition to BAT thermogenesis, providing the readers with a full understand of how NPFFR2 acts in the regulation of energy expenditure.*

Our response: Thank you for this comment. We agree that physical activity is an important contributor to energy expenditure (EE), and NPFFR2 may regulate EE via various pathways including altering physical activity, particular in female mice. While female *Npffr2*^{-/-} mice on chow exhibited some overall

increases in physical activity, the contribution to an increase in EE is only significant during the light phase, suggesting that the increased activity under this feeding condition only mildly contributes to the EE in these mice. However, under HFD conditions, female *Npffr2*^{-/-} mice show significantly decreased physical activity over the 24 h monitoring period and this coincides with the strong reduction in EE particularly during the dark phase, indicating physical activity as a mechanism for EE responding to a positive energy balance. We have now further examined the effects of diet on physical activity and how this may have impacted by NPFFR2 signalling by performing additional analyses, which are presented in **Fig 3k,l** in our revised manuscript.

Specifically we show that female WT mice increase physical activity on HFD versus chow, and that this was not seen in *Npffr2*^{-/-} mice. These data are consistent with observations in humans suggesting increased physical activity as one adaptive response to overfeeding or a positive energy balance (Apolzan et al., 2014), and further suggest that NPFFR2 signalling may play a role in the regulation of this response, lack of which may contribute to the impaired diet-induced thermogenesis and exacerbated diet-induced obesity in female *Npffr2*^{-/-} mice. Interestingly, in the male cohort, physical activity was comparable between WT and *Npffr2*^{-/-} mice under both chow and HFD conditions with no significant effects of HFD on physical activity in either genotype (**Fig 3k** in revised manuscript), suggesting a gender-specific role of NPFFR2 signalling in this regulation. These additional results are now added and discussed in our revised manuscript.

3) On page 11 lines 6-8, they described the results that HFD-induced increases in BAT temperature in WT mice were not seen in NPFFR2 knockout mice (Figure 4d), which actually showed a trend to decreased temperature. To me, this is likely due to a higher level of BAT temperature (more than 36°C) in this cohort of NPFFR2 knockout mice on chow compared to data in Figure 4a. Are there any environmental (such as stress) or body weight differences between these two tests?

Our response: The temperature data for **Fig. 4a** and the chow data in **Fig. 4c-d** were generated from the same cohort of mice, 6 weeks apart during which time the mice underwent a routine monitoring protocol which involved handling, weighing and body composition assessment (DEXA). In short, at 14 weeks of age mice were measured for basal BAT and Back temperatures, then measured again at 20 weeks of age, with data at these two time points presented in **Fig. 4a-b** and **4c-d**, respectively. While T_{BAT} , T_{Back} and $\Delta T_{BAT-Back}$ were comparable between WT and *Npffr2*^{-/-} mice at 14 weeks of age (**Fig 4a-b**), the 2nd measurement at 20 weeks showed higher T_{BAT} and T_{Back} in chow-fed *Npffr2*^{-/-} than WT mice with no significant difference in $\Delta T_{BAT-Back}$ between genotypes. These data represented in **Fig 4c-d** are re-graphed to show direct comparisons between genotypes in **new Supplementary Fig 4a-b** in our revised manuscript. Although we did not observe apparent stress-related behaviour and our behaviour testing did not show any anxiety phenotype (elevated plus maze and open field tests, **Supplementary Table1** in our revised manuscript), it is possible that *Npffr2*^{-/-} mice have a differential response to handling stress resulting in a higher T_{BAT} and T_{Back} in *Npffr2*^{-/-} mice. Importantly, since T_{BAT} and T_{Back} in *Npffr2*^{-/-} increased in parallel, $\Delta T_{BAT-Back}$ - a measurement of the direct contribution of BAT thermogenesis which is the focus of this study - was not significantly different between chow-fed *Npffr2*^{-/-} and WT mice either at 14 weeks or 20 weeks, nor was there any difference between the two time points within the same genotype (**new Supplementary Fig 4b**).

The comparable $\Delta T_{BAT-Back}$ between genotypes is consistent with our Western blotting data showing no significant difference in BAT UCP1 or PGC1 expression between WT and *Npffr2*^{-/-} mice under chow conditions. As such we represented the basal T_{BAT} and T_{Back} prior to any handling in **Fig 4a-b** to demonstrate the genotype effect, and since both the chow and HFD cohorts went through the same monitoring protocol (**Fig. 4a-b**), we separated the data collected 6 weeks later by genotype to demonstrate the diet effect on $\Delta T_{BAT-Back}$ and BAT thermogenesis (**Fig. 4c-d**).

Our data show that female *Npffr2*^{-/-} mice lack an increase in $\Delta T_{BAT-Back}$ in response to HFD (**Fig. 4c-d**), which we believe is due to an impaired diet-induced BAT thermogenesis rather than a higher T_{BAT} (greater than 36°C) in the chow group. Male NPFFR2^{-/-} mice under RT conditions had T_{BAT} lower than 36°C on chow and failed to increase $\Delta T_{BAT-Back}$ after 6 weeks of HFD feeding (**new Supplementary Fig. 4f, g**). In addition, *Npffr2*^{-/-} mice under TN had similar T_{BAT} to WT on chow and similarly failed to increase $\Delta T_{BAT-Back}$ in response to HFD feeding (**Fig. 4r, s**). Finally the lack of diet-induced increase in $\Delta T_{BAT-Back}$ in *Npffr2*^{-/-} mice is associated with a lack of increases in BAT UCP1 and PGC1a expression in

response to HFD, supporting an impaired BAT function and diet-induced BAT thermogenesis in mice lacking NPFFR2 signalling. We have added these results and discussions in our revised manuscript.

4) *The quality of Western blot image is low (Figure 4i), for instance, the loading control GAPDH was very weakly detected and the samples were unevenly loaded based on the levels of GAPDH. UCP-1 and PGC-1a levels should be normalized by the loading control GAPDH for group comparisons.*

Our response: We apologize for the low quality of the Western Blot, which is most likely due to the loss of resolution during conversion to pdf. In our Western experiment UCP1 and GAPDH were analysed from the same blot. Since the optimal exposure time for GAPDH is longer than that for UCP1, the image shown in our original manuscript, which was optimized for UCP1 exposure, consequently shows reduced GAPDH signal. We have provided an image of the same bands with longer exposure (for both UCP1 and GAPDH) in our revised manuscript. Importantly the data for UCP1 and PGC1 α protein expression presented as per mg protein (**Fig 4g, h**) and per BAT depot (**Fig 4j, k**) have already been normalized by the loading control GAPDH for all groups. The same analytical procedure have been used and published by Fischer AW et al. (Fischer et al., 2016). While most studies only express protein expression as per mg protein, we further presented data as per BAT depot since the size and protein content of the BAT is significantly different between genotypes and treatment groups (**Fig 4f**) which will impact on the total availability of UCP1 and PGC1 α protein in the BAT tissue and thus BAT thermogenic capacity. We have added more details and references in the Methods section regarding to Western analysis in our revised manuscript.

5) *On page 14 lines 9-11, they stated that “The POMC mRNA expression in the ARC was significantly increased in response to HFD in both WT and NPFFR2 knockout mice without significant differences between genotypes (Fig. 5e)”. However, the figure shows that ARC POMC mRNA expression was significantly increased only in NPFFR2 knockout mice on both diets. Please clarify.*

Our response: We apologize for this mix up. The reviewer is correct *Pomc* mRNA was only increased in the NPFFR2 deficient mice. This has now been corrected in the text.

6) *In the study of NPFFR2 deletion specifically in NPY neurons, they only showed the effect of this deletion on oxygen consumption in Figure 7. It would be better to present energy expenditure data as shown in Figure 3 for comparisons. Also, it would be better to specify whether the results were obtained from male mice, females or both.*

Our response: The data set of the conditional NPFFR2 receptor in NPY neurons has been expanded and presented in **new Fig. 8 and Fig. 9** in our revised manuscript. The results are from male mice to avoid potential side effects of tamoxifen treatment, which is needed for the induction of gene deletion, in female mice.

7) *Last, they report that “lack of NPFFR2 signaling on NPY neurons caused significant increases in femur length, femur BMD and a trend to an increase in femur BMC (Fig. 7h-j) compared to control mice”, but not provide the gender information. For me, this information is important because of the gender-dimorphic effect of global NPFFR2 knockout on bone mass, i.e., decreased femoral length in males (Fig. 6a) and unaltered femoral length in females (Fig. 6e). Particularly, these results from global knockout are not consistent with the finding of increased femoral length in mice with NPFFR2 deletion in NPY neurons. Thus, any similarities or discrepancies between NPY-neuron-specific and global NPFFR2 knockout need to be addressed and discussed.*

Our response: The results presented on the bone phenotype of conditional NPFFR2 knockout mice are again from male mice. We have expanded this analysis and now present the full set of bone data from the conditional NPY neuron-specific knockout mice in **new Fig. 9** of the revised manuscript, and discuss the phenotype of the conditional NPFFR2 knockout mouse models in more detail.

Reviewer #2 (Remarks to the Author):

General comments:

The authors of this paper explore the roles of the neuropeptide FF receptor 2 (NPFFR2) in the regulation of body weight (adiposity, energy expenditure, activity, food intake) and bone growth. They conclude that lack of NPFFR2 results in body-weight gain in mice fed a high-fat diet due to decreased energy expenditure as a result of decreased activation of brown fat thermogenesis. They provide some evidence that NPFF acting directly on NPY-expressing neurons is responsible for the phenotype of the knockout mice. The paper includes a wealth of data that, in general, are clearly described.

We thank the reviewer for the positive comments on our study.

Nevertheless, there are a number of problems that should be addressed.
1. If the thesis is that NPFF-expressing neurons in the NTS (sub-AP) project axons to the arcuate nucleus to act on NPFFR2 receptors on NPY-expressing neurons, then including experiments showing this connectivity is important. Showing that NPFFR2 is co-expressed with NPY in the Arc is a start, but is NPFF normally released in the vicinity of NPY neurons in the Arc?

Our response: We appreciate the comment by the reviewer. While NPFF is the most effective ligand for the NPFFR2, there is also considerable cross-reactivity within the RFamide peptide family. Thus NPFF, NPVF, PrRP and kisspeptin are all able to activate NPFFR2, although with different affinity and efficacy. Our results showing marked impairment of diet-induced thermogenesis and exacerbated diet-induced obesity suggesting that NPFFR2 signalling plays an important role in the regulation of energy homeostasis particularly during a positive energy balance. It is this centre position for NPFFR2 to mediate signalling pathways of several ligands, which we show have altered expression depending on energy status (**Fig. 1**) that intrigued us to study NPFFR2.

While we agree that the connectivity of the ligands to NPFFR2 and whether the effect of NPFFR2 on energy homeostasis results from a single ligand/receptor interaction or a combined effect of several ligands are important questions, they are outside the scope of this study. However, there is anatomical and biological evidence consistent with the notion that various ligands mediate physiological functions through Arc NPFFR2 pathways. For example NPVF fibers from the DMH have direct projections to Arc (Poling et al., 2013). Kisspeptin mRNA and fibers are present in the Arc and can activate Arc neurons via NPFFR2 signalling (Fu and van den Pol, 2010; Liu and Herbison, 2015). With regards to NPFF, which is predominantly expressed in the SubP of the brain stem, we conducted additional retro-tracing studies where red fluorescent retro-beads were unilaterally injected into the arcuate nucleus (Arc) of WT mice and brains were collected 2 weeks later for analysis. Importantly, in the brain stem region we observed specific red fluorescence-labelled cells and fibers in the area postrema (AP), SubP, and NTS, showing the direct input from these regions to the Arc. These data – presented in **new Supplementary Fig. 5** of our revised manuscript – support a direct projection of NPFF from SubP to the Arc, although the neuronal specificity for this projection requires future investigation. In addition, NPFF is also found in cerebrospinal fluid and plasma thus has easy access to the Arc and it has been suggested that NPFF can also act on NPFFR2 in a hormone-like fashion (Burlet-Schiltz et al., 2002; Sundblom et al., 1997). Furthermore, a recent study further demonstrated that plasma NPFF levels decrease in diet-induced obese mice and obese human, and increase in rodents and humans that were on caloric restriction (Waqas et al., 2017), consistent with our findings that SubP *Npff* mRNA expression shows significant decrease and increase in response to HFD and fasting, respectively (**Fig 1a-c**). We have added these discussion points to our revised manuscript.

2. It is unclear why the authors focused on NPFFR2 rather than NPFFR1 (or both). Is NPFFR1 also expressed in NPY neurons in the Arc? Is NPFFR1 activated by the same ligands as NPFFR2? Could inactivation of NPFFR2 be compensated for by NPFFR1? Does Kiss1 (made by neighboring neurons) also regulate NPFFR2 like it does NPFFR1?

Our response: As outlined in the introduction, the NPFFR2 is the predominate receptor expressed in the Arc hypothalamic nucleus with almost no detectable levels of NPFFR1 (Gouarderes et al., 2004). Following on from our response to question #1, we are mainly interested in NPFFR2 because of its ability to be acted upon by several RFamide family peptides (Elhabazi et al., 2013) and this indicates to us that the physiological functions from NPFFR2 signalling may be of critical importance. The endogenous ligands for NPFFR1 and NPFFR2 can cross-react in pharmacological tests with affinity and efficacy of the ligands being greater towards their own cognate receptor (Elhabazi et al., 2013). Regarding kisspeptins, *Kiss1* products have affinity towards NPFFR1 which is less than that to NPFFR2 (Oishi et al., 2011). In addition, whereas kisspeptins have affinity and efficacy towards NPFFR2, this cross-reactivity is uni-directional since NPFF receptor ligands have little activity towards GPR54 (Lyubimov et al., 2010; Oishi et al., 2011).

With regards to the co-expression of *Npffr1* and *Gpr54* in Arc NPY neurons, Arc single cell sequencing data from Henry *et al.* (Henry et al., 2015) show little expression of *Npffr1* or *Gpr54* in the Agrp/NPY neuronal population with approx. 0.8 and 0.2 fragments per kilobase of transcript per million mapped reads (FPKM) for *Npffr1* and *Gpr54*, respectively, where a FPKM value of less than 1 is generally considered as non-specific. In contrast *Npffr2* expression is high (approx. 6 FPKM per Arc Agrp/NPY cell). Please see graph below based on data from Henry *et al.* (Henry et al., 2015).

data based on Henry et al. (Henry et al., 2015).

To strengthen our conclusions, we performed additional qPCR to examine whether lack of *Npffr2* could be compensated for by *Npffr1* and whether this also leads to changes in the expression of *Gpr54* in *Npffr2* KO mice. Importantly, the expression of *Npffr1* in the hypothalamus was non-detectable in both WT and *Npffr2*^{-/-} mice after 40 cycles of PCR, consistent with a previous report (Gouarderes et al., 2004). There was also no significant change in the hypothalamic expression of *Gpr54* (**new Supplementary Fig 6a**). In addition, high fat diet feeding had no significant effects on the hypothalamic expression of *Gpr54* (**new Supplementary Fig 6b**), consistent with previous report (Luque et al., 2007). From this we conclude that the physiological effects we observe are the direct consequence of lack of NPFFR2 signalling and are not due to compensatory changes in expression of related receptors.

All together these data convincingly demonstrate the dominant expression of *Npffr2* in Arc neurons relative to other related receptors, and support an important action of NPFFR2 signalling towards Arc NPY neurons.

3. Does NPFF injection into the Arc of WT and NPFFR2 KO (or the conditional KO) mice reduce food intake by hungry mice.

Our response: Since feeding regulation by the NPFFR2 is not the main focus of this study we have not conducted such studies. From literature data it is known that i.c.v. injection of NPFF reduces fasting induced food intake (Bechtold and Luckman, 2006; Murase et al., 1996), however, due to the non-specific action of such a delivery, it is not clear which receptor mediates such a response. Importantly, from our results of the conditional NPY neuron-specific knockout model, where we do not see any difference in feeding (**Fig 8e**), we can conclude that direct signalling of NPFFR2 on NPY neurons activated by any of the RFamide ligands does not contribute to this behaviour.

4. How efficient is the conditional knockout? How long does it take for inactivation to occur?

Our response: Similar to results we have obtained using this inducible NPY^{cre} line in crosses with other floxed lines, the inactivation of the *Npffr2* gene occurs in less than 2 weeks after tamoxifen treatment and is more than 90% complete by then.

5. It is clear that mice can compensate for the loss of NPY/AgRP neurons, hence it follows that they can compensate for loss of any critical signaling molecule made by those neurons. Thus, the conditional knockout of NPFFR2 in adult mice is an important experiment. However, to conclude that NPY neurons mediate most of the effects of the constitutive knockout, the critical experiments shown in Fig. 2-4 should be replicated with the conditional KO. The time from when tamoxifen was added to the chow until experiments were performed is not stated; it should match that of the measurements in Fig. 2. The gender of the mice studied in Fig. 7 should be mentioned. What is the effect of the conditional knockout on BAT temperature, food intake, adiposity, insulin, leptin, etc.

Our response: The requested additional information has been added in **new Fig 8 and 9** in our revised manuscript.

6. *It seems odd that with constitutive knockout the major effect of NPFFR2 inactivation on energy expenditure is at night, whereas with the conditional inactivation it is during the day. What is the explanation for this.*

Our response: It is not unusual that germline and conditional adult onset deletions show differences in phenotype (Xu et al., 2018). The reason for this to happen one can only speculate. Most likely it is due to the selective nature of the deletion of NPFFR2 specifically in NPY neurons, which fulfil this specific function, while other NPFFR2-expressing non-NPY neurons have different functions. It is also possible that NPFFR2 signalling towards non-NPY neurons may control NPY neuronal function in an indirect way, which is not affected by NPY-neuron specific NPFFR2 deletion compared to global deletion. This has now been discussed in more detail in the revised manuscript.

7. *NPY is expressed broadly in the brain, so it is not clear that the conditional knockout of NPFFR2 in NPY neurons is having its effect primarily on the arcuate population. Conditional knockout in AgRP neurons would be more meaningful.*

Our response: We agree that NPY is widely expressed in the central nervous system, however, its highest expression is found in the Arc. On the other hand *Npffr2* mRNA expression is relatively limited with hypothalamus among the most highly expressed regions (Bonini et al., 2000; Liu et al., 2001). Nevertheless the amygdala - a region also involved in energy homeostatic control - expresses *Npy* and has moderate expression of *Npffr2* (Bonini et al., 2000; Cai et al., 2014; Douglass et al., 2017) (McGuire et al., 2011). Thus we compared NPFFR2 expression in hypothalamic NPY and amygdala NPY neurons using TRAP technology and found only a significant enrichment of *Npffr2* transcripts in the immunoprecipitated RNA of the NPY-TRAP mice in the hypothalamic fraction (consistent with our RNAscope data in **Fig 7d**) and not in the amygdala which actually shows a derichment demonstrating that NPFFR2 does not co-express with NPY in the amygdala. These results are now presented in **new Supplementary Fig. 7c**.

With regards to the use of an AgRP^{cre} driver line, it is important to note that while AgRP specific deletion is a possibility, it would miss a set of Arc NPY neurons since not all Arc NPY neurons (app 20%) contain AgRP (Luquet et al., 2005) (also unpublished data from us). As such we believe our conditional NPFFR2 knockout in NPY neurons is a more meaningful and a better model to study the action of NPFFR2 signalling in Arc NPY neurons in relation to the regulation of energy homeostasis.

8. *The experiment in Fig. 7d should be quantified. How much is pCREB reduced in NPY neurons? It looks like pCREB was reduced in non-NPY neurons in the Arc as well; that should also be quantified. Overall, the observation that NPFFR2 regulates thermogenesis, which affects energy expenditure and*

body weight is interesting. A paper more focused on the circuitry involved would be appreciated. Some of the other data (e.g. bone growth) detracts from these central questions.

Our response: The pCREB data have now been analysed in greater detail and the quantifications are presented. Whereas having a similar number of NPY neurons in the Arc, the Forskolin+NPFF group had significantly reduced NPY neurons (~20%) co-localized with pCREB compared to the Forskolin group. The pCREB-ir positive neurons in the Arc showed a trend towards a decrease in the Forskolin+NPFF group compared to the Forskolin group, although this was not statistically significant. These data have been added in **new Fig. 7f and Supplementary Table 2** in our revised manuscript.

With regards to the bone data, this reviewer expresses the opposite view to reviewer#1. We believe that the bone data are of importance based on the fact that Arc NPY is a major controller of bone mass (Baldock et al., 2014; Baldock et al., 2009) and thus Arc NPY might be the special target via which NPFFR2 signalling exerts a coordinated control over energy-demanding processes including BAT thermogenesis and bone formation. We have now discussed this possibility more extensively.

Editorial comments:

1. *Approved mouse gene names in italics should be used when referring to genes and their mRNAs. All mouse gene names start with a capital letter followed by lowercase letters and numbers. Ob/ob is not a gene name, it is an allele of Lep gene.*
2. *Authors should consistently use hyphens when using nouns as adjectives: a few examples, high-fat diet, neuropeptide-expressing neurons, ad libitum-fed mice, activity related alterations, metabolism-related phenotypes, tail-flick test. p 11, No hyphen "after 6 weeks of chow"*
3. *Always leave a space between numbers and their units, e.g. 24 h (especially bad in Methods)*
4. *Use em-dash appropriately (not an comma and a hyphen)*
5. *The genetic background of mice should be indicated.*
6. *Approval of animal experiments should be included.*
7. *P 6, "8 weeks of feeding" no hyphen, spelling*
8. *P 6, changes.....suggest (not suggests)*
9. *P 13, "these results show..." not showing*
10. *ISH does not need to be in italics.*
11. *P 17, NPFFR2 lox/lox, flox/flox, flox/fox; what is the distinction between these different designations?*
12. *P 19, Is "overwriting" meant or should this be overriding?*
13. *P 21, colocalization (spelling)*

Thank you for pointing this editorial issues out, they have all been attended too.

Reviewer #3 (Remarks to the Author):

This study evaluates the role of neuropeptide FF receptor 2 (NPFFR2) in energy balance. The manuscript suggests that NPFFR2 plays a critical role in NPY neurons to control energy expenditure and BAT activity particularly when mice are fed a HFD. Although the hypothesis is interesting, there are important issues on this study:

1. *Figure 5. These results demonstrate that NPFFR2 signalling regulates not only Arc NPY levels but also POMC levels. It is surprising that the authors just discard the potential role of NPFFR2 signalling on POMC neurons, which are also relevant in the control of thermogenesis and whole body energy homeostasis. This aspect is of particular relevance since there are also NPY neurons that do not express NPFFR2 (Fig. 7C-ii) as well as non-NPY neurons that do express NPFFR2 (Fig. 7C-iii), and as the authors state, this suggests that NPFFR2- signalling could also affect other Arc neuronal pathways. Therefore, to do a full characterization of the role of NPFFR2- signaling in POMC neurons is important. Moreover, from this figure it is not possible to conclude that "through that influences the responsiveness of Arc NPY neurons to a HFD that subsequently results in impaired diet-induced BAT thermogenesis in NPFFR2-/- mice."*

Our response: We appreciate the comment by the reviewer and we apologize for not having this made more clear in the original manuscript. However literature clearly indicates that NPFFR2 signalling on POMC neurons is of no importance due to the fact that these receptors are expressed only in minute amounts on POMC neurons and there is no change of expression due to altered energy status. For example single cell transcriptome data from the hypothalamus generated by the Linnerarson lab (see below figure) (Romanov et al., 2017) (<http://linnerarssonlab.org/hypothalamus/>) clearly shows no co-localization of NPFFR2 and POMC (Highlighted in yellow).

Image generated from (<http://linnarssonlab.org/hypothalamus/>).

Consistent with the data above that *Npffr2* is predominantly expressed in the NPY neurons of the Arc, but not in the POMC neurons, we have extracted another set of published single cell RNA-sequencing data that has looked at the transcriptome profile of NPY and POMC neurons by using AgRP-GFP and POMC-GFP mice (Henry et al., 2015). We re-analysed the dataset and confirmed firstly that the expression of *Npy* and *Agrp*, but not *Pomc*, are highly expressed in NPY/AGRP neurons, and *Pomc* is highly expressed only in POMC neurons, but not *Npy* and *Agrp*. Importantly, *Npffr2* expression is also highly expressed specifically in NPY neurons compared to POMC neurons that only had an expression level of fragments per kilobase of transcript per million mapped reads (FPKM) value of less than 1, which is generally considered as nonspecific (See below figure).

Data generated by Henry et al., 2015 (Henry et al., 2015).

Nevertheless, we have now also performed double *in situ* hybridisation employing RNAscope and can confirm that while there is a consistent co-localisation of *Npffr2* with *Npy*, little *Npffr2* transcripts were detected to colocalise with *Pomc* neurons. This has now been added as **new Supplementary Fig. 7e**.

Therefore, as discussed in the original manuscript the up-regulation of *Pomc* mRNA in NPFFR2^{-/-} mice is most likely the consequence of the reduced NPY levels in these mice that reduce the inhibitory tone of NPY neurons onto POMC neurons leading the up-regulation of *Pomc* mRNA.

2. Another key aspect would be to measure the electrical activity of NPY (and POMC) neurons.

Our response: Thank you for this suggestion and while possible this is clearly outside the scope of this manuscript and would only add minor additional information to the overall conclusions of this manuscript.

3. NPFFR2 is expressed in other tissues besides the brain (i.e. adipose tissue macrophages, see PMID: 28581443) and seems to be metabolically important. Thus, studies using NPFFR2^{-/-} mice should be taken with caution here, and complete studies in conditional KO models are essential to provide convincing results on the specificity of hypothalamic NPFFR2

Our response: Thank you for this comment. While it is true that peripheral NPFFR2 receptors have been reported, their level of expression is rather low (Bonini et al., 2000; van Harmelen et al., 2010) and the tissue distribution (eg macrophages) (Waqas et al., 2017) makes them the less likely candidates to control Arc nucleus circuits and BAT functions.

However, we performed additional experiments and examined *Npffr2* expression in the BAT and subcutaneous white adipose tissue - the adipose tissue type that has the greatest beige capacity and thermogenic potential - by using quantitative RT-PCR. *Npffr2* expression in the BAT is non-detectable after 40 cycles of PCR. In the subcutaneous adipose tissue there is extremely low expression of *Npffr2* that is approx. 30-fold less than its hypothalamic expression. These results are presented in **new Fig. 5c** in our revised manuscript.

4. The characterization, or at least the description here, of mice lacking NPFFR2 in NPY neurons is very succinct. For example, data on food intake, body composition and temperature, which are from my

point of view essential in this study, are not shown. In addition, the phenotype of these mice does not entirely recapitulate the findings observed in the global KO. For instance, in conditional KOs, daily caloric intake is identical between genotypes when fed a HFD while it was different in global KO (Fig 2i), suggesting that the effects of NPFFR2 signaling on feeding may be independent of NPY, and this may take us again to POMC neurons.

Our response: See also our response to reviewer #2 point 6. A more extensive characterisation of the NPY neuron-specific KO model has been added in new Figs 8 and 9 and the results are discussed in more details in our revised manuscript.

5. To fully characterize the specificity of the results, over-expression in NPY and POMC neurons is advisable.

Our response: While a possibility, this type of experiments is clearly outside the scope of this study.

Minor:

Page 8. ...over a 10-week monitoring should be changed by 20-week?

Our response: The mice were monitored from 10 weeks of age to 20 weeks of age, thus it was a 10 week monitoring period. We have made this more clear in our revised manuscript.

Reviewer #4 (Remarks to the Author):

Zhang et al's manuscript describes a new signalling pathway involving NPFF receptor 2 (NPFFR2) that controls food intake and energy expenditure by promoting diet-induced thermogenesis in the brown adipose tissue (BAT). The authors provide convincing results that NPFFR2 is required to control energy expenditure and that the signaling pathway activating BAT thermogenesis is initiated in NPY neurons. The results are novel and of high interest in the field. The experimental approaches and the statistical analysis are of high standard. At the end of the manuscript they used Translating Ribosome Affinity Purification Approach to provide evidences that NPFFR2 signaling acts directly on NPY neurons. For this they generated mice that express a GFP translational fusion of the ribosomal protein L10 (L10a-GFP), refereed to as NPY-TRAP, by crossing TRAP lox/lox mice with NPY CRE/+ mice. The authors verified accumulation of L10a-GFP in the Arcuate nucleus by using fluorescent microscopy. Upon immunoprecipitation (IP) with antiGFP antibodies, the authors showed that L10a-GFP transcripts were present in the IPs of NPY-TRAP mice, but not in the Ips of Wt mice. These IPs were enriched in NPY mRNAs as compared with IP of WT mice. NPFFR2 was also found enriched in the IPs from NP-Y neurons.

At this point, it is my impression that it is important to show the mRNA levels of NP-Y, L10a-GFP, and NPFFR2 in IPs obtained from the parental TRAP lox/lox and NPY Cre/+ mice instead of in WT mice to straighten the conclusion of the experiment. To provide evidence of enrichment they can also compare IP of NPY-TRAP mice with IP from mice that ubiquitously express L10a-GFP. This is of particular importance since the authors later showed, by in situ hybridization, that there is juts partial co-expression of NP-Y and NPFFR2.

"...At this point, it is my impression that it is important to show the mRNA levels of NP-Y, L10a-GFP, and NPFFR2 in IPs obtained from the parental TRAP lox/lox and NPY Cre/+ mice instead of in WT mice to straighten the conclusion of the experiment."

Our response: We thank the reviewer for the positive comments and appreciation of the study, and also the suggestion of using an alternative mouse model that ubiquitously expresses the L10a-GFP as positive control. For that, we now provide a more comprehensive analysis of the TRAP results. Firstly, we used another mouse line (InsulinCre) that also activates the TRAP expression in the hypothalamus

but in a different subset of neurons than the one for NPY. Induction of TRAP by this line was confirmed in brains sections of induced and control mice (**new supplementary Fig. 7a**).

We dissected out the arcuate nucleus (Arc) of the Ins-TRAP mice in the same way as we do for our NPY-TRAP mice and performed TRAP with the same protocol. Interestingly, the expression of *Npy*, *Npffr2* and GFP in the NPY-TRAP, Ins-TRAP and WT-TRAP Input RNA was not different (See **Fig. 7a and 7b**; as well as **new supplementary Fig. 7b**), indicating that insertion of the transgene in these mice did not alter the baseline expression level of *Npy* and *Npffr2* in the Arc, and are therefore comparable to each other for this study.

Next, we confirmed that GFP was successfully enriched in both the NPY-TRAP and Ins-TRAP mice but not in the WT-TRAP mice, confirming that our immunoprecipitation protocol is highly efficient and valid (See **new supplementary Fig. 7b**). Importantly, while the enrichment of GFP transcript is not different in the NPY-TRAP mice from the Ins-TRAP mice, the enrichment of *Npy* and *Npffr2* was significantly higher in the NPY-TRAP mice compared to the Ins-TRAP mice (**new Fig 7b and 7c**), confirming that *Npffr2* is highly co-expressed with *Npy* in neurons of the Arc but not the ones induced by the insulin cre. This again is consistent with our RNAscope data showing that *Npffr2* is predominantly co-expressed with *Npy* in the Arc (**Fig. 7d**).

References

- Apolzan, J.W., Bray, G.A., Smith, S.R., de Jonge, L., Rood, J., Han, H., Redman, L.M., and Martin, C.K. (2014). Effects of weight gain induced by controlled overfeeding on physical activity. *Am J Physiol Endocrinol Metab* 307, E1030-1037.
- Baldock, P., Lin, S., Zhang, L., Karl, T., Shi, Y., Driessler, F., Zengin, A., Horner, B., Lee, N., Wong, I., et al. (2014). Neuropeptide Y Attenuates Stress-Induced Bone Loss Through Suppression of Noradrenaline Circuits. *J Bone Miner Res*.
- Baldock, P.A., Lee, N.J., Driessler, F., Lin, S., Allison, S., Stehrer, B., Lin, E.J., Zhang, L., Enriquez, R.F., Wong, I.P., et al. (2009). Neuropeptide Y knockout mice reveal a central role of NPY in the coordination of bone mass to body weight. *PLoS One* 4, e8415.
- Bechtold, D.A., and Luckman, S.M. (2006). Prolactin-releasing Peptide mediates cholecystokinin-induced satiety in mice. *Endocrinology* 147, 4723-4729.
- Bonini, J.A., Jones, K.A., Adham, N., Forray, C., Artymyshyn, R., Durkin, M.M., Smith, K.E., Tamm, J.A., Boteju, L.W., Lakhani, P.P., et al. (2000). Identification and characterization of two G protein-coupled receptors for neuropeptide FF. *J Biol Chem* 275, 39324-39331.
- Burlet-Schiltz, O., Mazarguil, H., Sol, J.C., Chaynes, P., Monsarrat, B., Zajac, J.M., and Roussin, A. (2002). Identification of neuropeptide FF-related peptides in human cerebrospinal fluid by mass spectrometry. *FEBS Lett* 532, 313-318.
- Cai, H., Haubensak, W., Anthony, T.E., and Anderson, D.J. (2014). Central amygdala PKC-delta(+) neurons mediate the influence of multiple anorexigenic signals. *Nat Neurosci* 17, 1240-1248.
- Douglass, A.M., Kucukdereli, H., Ponserre, M., Markovic, M., Grundemann, J., Strobel, C., Alcalá Morales, P.L., Conzelmann, K.K., Luthi, A., and Klein, R. (2017). Central amygdala circuits modulate food consumption through a positive-valence mechanism. *Nat Neurosci* 20, 1384-1394.
- Elhabazi, K., Humbert, J.P., Bertin, I., Schmitt, M., Bihel, F., Bourguignon, J.J., Bucher, B., Becker, J.A., Sorg, T., Meziane, H., et al. (2013). Endogenous mammalian RF-amide peptides, including PrRP, kisspeptin and 26RFa, modulate nociception and morphine analgesia via NPFF receptors. *Neuropharmacology* 75, 164-171.

Fischer, A.W., Hoefig, C.S., Abreu-Vieira, G., de Jong, J.M., Petrovic, N., Mittag, J., Cannon, B., and Nedergaard, J. (2016). Leptin Raises Defended Body Temperature without Activating Thermogenesis. *Cell Rep* 14, 1621-1631.

Fu, L.Y., and van den Pol, A.N. (2010). Kisspeptin directly excites anorexigenic proopiomelanocortin neurons but inhibits orexigenic neuropeptide Y cells by an indirect synaptic mechanism. *J Neurosci* 30, 10205-10219.

Gouarderes, C., Puget, A., and Zajac, J.M. (2004). Detailed distribution of neuropeptide FF receptors (NPFF1 and NPFF2) in the rat, mouse, octodon, rabbit, guinea pig, and marmoset monkey brains: a comparative autoradiographic study. *Synapse* 51, 249-269.

Henry, F.E., Sugino, K., Tozer, A., Branco, T., and Sternson, S.M. (2015). Cell type-specific transcriptomics of hypothalamic energy-sensing neuron responses to weight-loss. *Elife* 4, 09800.

Liu, Q., Guan, X.M., Martin, W.J., McDonald, T.P., Clements, M.K., Jiang, Q., Zeng, Z., Jacobson, M., Williams, D.L., Jr., Yu, H., et al. (2001). Identification and characterization of novel mammalian neuropeptide FF-like peptides that attenuate morphine-induced antinociception. *J Biol Chem* 276, 36961-36969.

Liu, X., and Herbison, A. (2015). Kisspeptin regulation of arcuate neuron excitability in kisspeptin receptor knockout mice. *Endocrinology* 156, 1815-1827.

Luque, R.M., Kineman, R.D., and Tena-Sempere, M. (2007). Regulation of hypothalamic expression of KiSS-1 and GPR54 genes by metabolic factors: analyses using mouse models and a cell line. *Endocrinology* 148, 4601-4611.

Luquet, S., Perez, F.A., Hnasko, T.S., and Palmiter, R.D. (2005). NPY/AgRP neurons are essential for feeding in adult mice but can be ablated in neonates. *Science* 310, 683-685.

Lyubimov, Y., Engstrom, M., Wurster, S., Savola, J.M., Korpi, E.R., and Panula, P. (2010). Human kisspeptins activate neuropeptide FF2 receptor. *Neuroscience* 170, 117-122.

McGuire, J.L., Larke, L.E., Sallee, F.R., Herman, J.P., and Sah, R. (2011). Differential Regulation of Neuropeptide Y in the Amygdala and Prefrontal Cortex during Recovery from Chronic Variable Stress. *Front Behav Neurosci* 5, 54.

Murase, T., Arima, H., Kondo, K., and Oiso, Y. (1996). Neuropeptide FF reduces food intake in rats. *Peptides* 17, 353-354.

Oishi, S., Misu, R., Tomita, K., Setsuda, S., Masuda, R., Ohno, H., Naniwa, Y., Ieda, N., Inoue, N., Ohkura, S., et al. (2011). Activation of Neuropeptide FF Receptors by Kisspeptin Receptor Ligands. *ACS Med Chem Lett* 2, 53-57.

Poling, M.C., Quennell, J.H., Anderson, G.M., and Kauffman, A.S. (2013). Kisspeptin neurones do not directly signal to RFRP-3 neurones but RFRP-3 may directly modulate a subset of hypothalamic kisspeptin cells in mice. *J Neuroendocrinol* 25, 876-886.

Romanov, R.A., Zeisel, A., Bakker, J., Girach, F., Hellysaz, A., Tomer, R., Alpar, A., Mulder, J., Clotman, F., Keimpema, E., et al. (2017). Molecular interrogation of hypothalamic organization reveals distinct dopamine neuronal subtypes. *Nat Neurosci* 20, 176-188.

Sundblom, D.M., Kalso, E., Tigerstedt, I., Wahlbeck, K., Panula, P., and Fyhrquist, F. (1997). Neuropeptide FF-like immunoreactivity in human cerebrospinal fluid of chronic pain patients and healthy controls. *Peptides* 18, 923-927.

van Harmelen, V., Dicker, A., Sjolín, E., Blomqvist, L., Wiren, M., Hoffstedt, J., Ryden, M., and Arner, P. (2010). Effects of pain controlling neuropeptides on human fat cell lipolysis. *Int J Obes (Lond)* 34, 1333-1340.

Waqas, S.F.H., Hoang, A.C., Lin, Y.T., Ampem, G., Azegrouz, H., Balogh, L., Thuroczy, J., Chen, J.C., Gerling, I.C., Nam, S., et al. (2017). Neuropeptide FF increases M2 activation and self-renewal of adipose tissue macrophages. *J Clin Invest* 127, 2842-2854.

Xu, J., Bartolome, C.L., Low, C.S., Yi, X., Chien, C.H., Wang, P., and Kong, D. (2018). Genetic identification of leptin neural circuits in energy and glucose homeostases. *Nature* 556, 505-509.

Point by point response to reviewers comments

Reviewer #1 (Remarks to the Author):

The authors have addressed my concerns extensively in this revision.

We thank the reviewer for the appreciation on our efforts to improve the manuscript.

Reviewer #2 (Remarks to the Author):

I have noted a number of editorial suggestions below:

Line 37, cold-induced thermogenesis (add hyphen)

Line 37 Npffr2. Authors should be consistent in using either all capital letters (e.g. line 33) or not for this receptor

Line 39 “downstream PVN-TH neuronal BAT controlling pathways”. This is a complicated modifier of pathways with many abbreviations/concepts that are not defined. Perhaps better to say, “downstream pathways that control BAT thermogenesis” and leave details of circuitry for Discussion

Line 40, “in this tissue” is ambiguous, consider: “in BAT”

Line 43, “and bone tissue” There is no mention of bone anywhere else in Abstract; thus, this last phrase doesn't fit well.

Line 51, positive-energy balance (hyphen), also lines 167, 182, 684

Line 54, humans (plural)

Line 55 define UCP1

Line 56, creatine-futile-cycle pathway (add 2 hyphens)

Line 81, “the primarily expressed hypothalamic-26RF” Either “remove the primarily expressed hypothalamic-” 26RF, or say “26RF that is primarily expressed in the hypothalamus....”

Line 108, Adding “In order” to a sentence has no effect on meaning of sentence and could be eliminated. See also lines 324, 343

Line 109, define WT

Line 134, I would leave it up to the reader to determine wither the effects are “profound” or not. The 2-fold effects may not be biologically significant. Mice heterozygous for neuropeptide genes generally have no phenotype compared to WT mice.

Line 144, “breeding of heterozygous Npffr2+/- mice (with each other).....” then leave out “from heterozygous Npffr2+/- 146 breeding pairs” later in sentence

Line 149 “was not affecting” consider “did not affect”

Line 155, “with regard to” not “regards”

Line 157, open-field test, elevated-plus-maze test (hyphens)

Line 216, It is not clear from text or Figure legend 3, when EE was measured relative to time on the HFD. Comment on effect of body-weight difference between groups would be useful.

Line 255, “thermo” is not a word. Consider, “thermogenic”

Line 343, add “genes” after Npvf and Kiss1

Line 343, Consider changing sentence: To investigate whether NPF, the other major ligand of NPFFR2 that is found most prominently in the SubP, may also interact with Arc neurons, we...

**Lines 353-354, This is a weak conclusion. “These results show that neurons in the AP and SubP regions have direct projections to the Arc, supporting the notion that Npff-expressing neurons in the SubP may act on Arc Npffr2-expressing neurons via a direct projection.” This experiment only shows connectivity between the two brain regions but says nothing about possibility the Npff from NTS/SubP region may innervate neurons in the ARC. Authors should do in situ hybridization to colocalize red beads with NPff mRNA in NTS/SubP region. This should either be eliminated or expanded as suggested. See also line 650-653.

Line 358, The approved mouse gene name for Gpr54 is *Kiss1r* (italics), see also line 873

Line 394, thyrotropin and oxytocin do not need to be capitalized in this context

Line 399, The experiments described to not measure the responsiveness of PVN TH neurons. They only show that Th mRNA changes.

Line 438-439 change hyphens to commas to set off the comment “a region expressing”

Line 445, single-cell sequencing (hyphen)

Line 448, “other Arc neurons” leave out “pathways”

Lines 462 and 463, use m dashes rather than hyphens. Likewise, lines 503-504, 632-633

Line 481, Consider changing to: “no significant difference in the data collected from these two lines”

Line 488, regard (not regards), See also lines 512, 559

Line 546, Use the abbreviation for high fat diet

Line 553, TH-positive neurons (hyphen). Add reference for role of TH neurons to this sentence.

Line 641, NPFFR2-dependent (add hyphen)

Line 692, “resulting a favour towards...” Consider: shifting the balance towards...

Line 695, Constitutive lack of NPY does not lead to obesity

Line 731, tamoxifen does not need to be capitalized

Lines 711-731, indicate the genetic background of the mice used for these experiments.

Line 861 & 875, The approved mouse gene name is *Actb* (italics)

We thank the reviewer for pointing out these editorial issues which have been all attended to. We have also clarified some of the statements in line with the reviewer's suggestions.

Reviewer #3 (Remarks to the Author):

The rebuttal as well as the new data convincingly demonstrate that *Npffr2* is colocalized with NPY. However, this reviewer still believes that it is important to know whether *Npffr2* can modulate the activity of NPY neurons –especially in studies aimed to be published in outstanding journals-. In line with this, an experiment over-expressing *Npffr2* in NPY neurons is equally important and I do not understand why these experiments are outside the scope of this study.

We are pleased that the reviewer now agrees that we have convincingly demonstrated that NPFFR2 is colocalised with NPY, which implies that signalling through NPFFR2 will influence NPY neuronal activity and function. However, we respectfully disagree that additional experiments are required to show this again. The various gene ablation models demonstrate that lack of NPFFR2 alters NPY mRNA levels (Fig. 5I). Furthermore, the evaluation of alterations in pCREB activity in NPY neurons upon NPFF initiated NPFFR2

signalling (Fig 7e) is further proof of a direct action mediated by NPF2R on NPY neuronal activity. The basic characterization of NPF2R, including its electrochemical properties, have been published before (Bonini JA, et al; J Biol Chem. 2000 Dec 15;275(50):39324-31) and as such a repeat of that would not add any vital additional information to the already extensive dissection of the interaction of NPF2R signalling and NPY function in the control of diet-induced adaptive thermogenesis.

Secondly, in the case of performing experiments with specific overexpression of NPF2R in Arc NPY neurons, the reviewer has overlooked the fact that not all Arc NPY neurons express the NPF2R receptor. While models that selectively delete the NPF2R in NPY neurons, as we show are very useful, the over production of the NPF2R in these neurons is not. Using NPY-Cre as the driver for the NPF2R deletion will eliminate these receptors from the genome in all NPY neurons. However, only in NPY neurons that endogenously express NPF2R this will have a functional consequence, but it will have no consequence in NPY neurons that do not transcribe the NPF2R. In contrast, if one would uses an AAV-FLEX viral vector to specifically re-introduce or over-express the NPF2R in Arc NPY neurons (as the reviewer suggests) this would lead to the ectopic expression of the NPF2R in all NPY neurons – thereby creating functional artefacts to the system. The outcome of such an experiment where NPF2R receptors will be expressed in every NPY neuron would rather confuse the issue than adding any useful data. As such this suggested experiment is not only out of the scope of this study but would also only produce meaningless data.

Reviewer #4 (Remarks to the Author):

The authors have nicely and satisfactorily addressed the issue raised to previous version of the manuscript concerning the TRAP experiments by including a mouse line (InsulinCre) that activates TRAP expression in the hypothalamus, but in a different subset of neurons than the one for NPY. They verified expression of GFP in these neurons (Figure 7a and Supplemental Figure 7a) and

also showed similar enrichment of GFP in the immunoprecipitates isolated from NPY-TRAP and Ins-TRAP mice, but not from the WT-TRAP mice (Supplemental Figure 7b), demonstrating the success of TRAP experiments in both NPY-TRAP and Ins-TRAP mice. More importantly, TRAP experiments followed by RT-qPCR confirmed that the enrichment of *Npy* and *Npffr2* was higher in the IPs obtained from NPY-TRAP mice than from those of Ins-TRAP mice (Figure 7b and 7c).

The new experiments and information presented by the authors certainly strengthen the conclusions of TRAP experiments.

We also thank this reviewer for acknowledging our efforts in improving the manuscript.